# Long-term monitoring of intravital biological processes using fluorescent protein-assisted NIR-II imaging

Muxiong Chen[1,2,4], Zhe Feng [3,4], Xiaoxiao Fan[3,4], Jun Sun[1], Weihang Geng[3], Tianxiang Wu[3], Jinghao Sheng[1,2] ✉, Jun Qian [3] ✉ & Zhengping Xu [1,2] ✉

High spatial resolution, low background, and deep tissue penetration have made near-infrared II (NIR-II) fluorescence imaging one of the most critical tools for in vivo observation and measurement. However, the relatively short retention time and potential toxicity of synthetic NIR-II fluorophores limit their long-term application. Here, we report the use of infrared fluorescent proteins (iRFPs) as in vitro and in vivo NIR-II probes permitting prolonged continuous imaging (up to 15 months). As a representative example, iRFP713 is knocked into the mouse genome to generate a transgenic model to allow temporal and/ or spatial expression control of the probe. To demonstrate its feasibility in a genuine diagnostic context, we adopt two liver regeneration models and successfully track the process for a week. The performance and monitoring efficacy are comparable to those of μCT and superior to those of indocyanine green dye. We are also able to effectively observe the pancreas, despite its deep location, under both physiological and pathological conditions. These results indicate that the iRFP-assisted NIR-II fluorescence system is suitable for monitoring various tissues and in vivo biological processes, providing a powerful noninvasive long-term imaging platform.

Fluorescence imaging is one of the most widely used techniques for in vivo structural and functional studies[1]. Imaging using the second near-infrared region (NIR-II, 900–1880 nm) enables direct visualization and real-time monitoring of deep biological structures and processes with an impressive degree of clarity[2–4]. Its potential greatly exceeds that obtained using either the visible (360–760 nm) or NIR-I (760–900 nm) windows due to suppressed imaging background resulting from moderate tissue absorption and relatively low tissue scattering, minimized autofluorescence, and negligible phototoxicity to living cells[5]. To achieve deep-tissue optical imaging with a higher signal-to-background ratio and better spatial resolution, the key challenge is to develop fluorescent probes with bright NIR-II emissions[6].

To date, rare-earth doped nanoparticles (RENPs)[7–10], quantum dots (QDs)[11–14], and certain organic fluorescent dyes[15–18] have been developed for use as NIR-II fluorescent probes in multifunctional biomedical imaging. However, because they are chemically synthesized, these probes suffer from serious limitations, including that they are nonrenewable in living systems and can present potential risks of biotoxicity. During cell division, the partition of such probes between the two daughter cells will result in signal attenuation and lead to the probe being no longer detectable after several cell duplications[19]. Moreover, synthetic probes can be rapidly broken down or eliminated by biological processes in the body[20]. In addition, depending on the type, exogenous fluorescent probes may exhibit certain levels of

[1]Institute of Environmental Medicine and Department of General Surgery, Sir Run Run Shaw Hospital, Zhejiang University School of Medicine, Hangzhou, China. [2]Liangzhu Laboratory, Zhejiang University Medical Center, Hangzhou, China. [3]State Key Laboratory of Modern Optical Instrumentations, Centre for Optical and Electromagnetic Research, College of Optical Science and Engineering, International Research Center for Advanced Photonics, Zhejiang University, Hangzhou, Zhejiang, China. [4]These authors contributed equally: Muxiong Chen, Zhe Feng, Xiaoxiao Fan. ✉e-mail: jhsheng@zju.edu.cn; qianjun@zju.edu.cn; zpxu@zju.edu.cn

cellular toxicity[21]. Therefore, these probes are not practical for use in prolonged imaging. As an alternative and more biocompatible option, fluorescent proteins (FPs) have great advantages over chemically synthesized probes in long-term monitoring. This is because FPs can be genetically encoded into cells and passed onto daughter cells[22]. As this occurs, the fluorescence signal intensity will be enhanced, rather than dissipated, as cells proliferate. To date, visible light-emitting FPs have been commonly used for in vitro imaging. However, these kinds of FPs do not provide sufficient performance for in vivo macro-observation, mainly due to the serious levels of tissue attenuation. Therefore, near-infrared fluorescent proteins (NIR FPs) with longer characteristic emission wavelengths are in high demand.

Several lines of NIR FPs have been reported[23, 24]. Among them, an infrared fluorescent protein (iRFP) series, including iRFP670, iRFP682, iRFP702, iRFP713, and iRFP720, has emission and excitation spectra at near-infrared wavelengths[25]. iRFP is an engineered version of the bacteriophytochrome from *Rhodopseudomonas palustris* and enables noninvasive and long-term in vivo visualization[23–29]. However, thus far, only the NIR-I region has been exploited.

To explore the application of iRFPs in the NIR-II region, we first characterized the spectral properties of this probe and found that its emission tail extended into the 900–1300 nm region and exhibited a high level of brightness. The in vitro and in vivo observation data indicated that iRFPs could serve as fluorescent protein probes for long-duration NIR-II imaging (at least 15 months). Furthermore, to achieve tissue-specific and controlled expression of this probe, we generated inducible iRFP713 transgenic mice with gene-knockin and Cre-LoxP technologies and found that the use of this protein probe could not only enable clear observation of the liver and its regeneration process but also image the pancreas, a deeply located organ, under both physiological and pathological conditions. Overall, this study provides proof of concept that NIR-II bioimaging of iRFP713 via off-peak fluorescence emission has high temporal and spatial resolution, offers long-term monitoring capability, and may serve as a powerful platform for the imaging of various internal biological processes.

## Results

### Characterization and in vitro imaging of iRFP in the NIR-II region
To characterize of iRFP in the NIR-II region, we first purified all five reported iRFPs using protein expression and purification procedures (Supplementary Fig. 1a, b) and then measured their spectral properties (Fig. 1a, b). Notably, although the iRFPs showed maximum emissions from 670 to 720 nm in the visible and NIR-I regions, they also exhibited spectral tails from 900–1300 nm, indicating that their emission spectra extend into the NIR-II region (Fig. 1c). These results indicated the potential of iRFPs to be utilized as fluorescent probes for NIR-II imaging.

To further explore this possibility, we first measured the brightness of each protein in vitro. When excited with a 690 nm continuous wave (CW) laser with a 900 nm longpass (LP) filter, the five iRFP solutions exhibited various degrees of fluorescence. It should be noted that the iRFPs possess high molar absorption coefficients[26]. Thus, they showed good fluorescence performance, as shown in Fig. 1d, even though the light absorption proportion at 690 nm of some proteins was relatively low, as shown in the normalized absorption spectra in Fig. 1a. For example, we tested the absolute QY of iRFP713 using an absolute PL quantum yield spectrometer and calculated the NIR-II QY (beyond 900 nm) as -0.33%. Although the QY is not very high, the molar absorption coefficient of iRFP713 protein at the main peak is as high as 98000 $M^{-1}cm^{-1}$ [26], leading to a strong NIR-II emission. As iRFP713 displayed the brightest relative fluorescence intensity (Fig. 1d and Supplementary Fig. 1c), this probe was selected for further research.

To test the applicability of iRFP713 in NIR-II imaging of mammalian cells, we constructed three iRFP713-expressing stable cell lines and subsequently analyzed their intracellular fluorescence using NIR-II wide-field microscopy. Our data showed that iRFP713 fluorescence could be continuously detected in all the stable cells (Supplementary Fig. 2a), and the characteristic NIR-II fluorescence remained unchanged in the presence or absence of external biliverdin (BV), suggesting that this fluorescent probe does not need a cofactor to function (Supplementary Fig. 2b).

Adequate penetration has always been a challenge for fluorescence imaging in vivo but remains of vital importance for deciphering biological structures located deeper within the body. To evaluate the tissue penetration ability of iRFP713, we performed intralipid phantom imaging using a capillary tube containing iRFP713 immersed in a 1% intralipid solution at the indicated phantom depths (Fig. 1e). As shown in Fig. 1f, the tube boundary could be clearly distinguished in the NIR-II region (>900 nm), even at a 6 mm immersion depth, while it was blurred and invisible in the NIR-I region (800–900 nm). The quantitative results of the coefficient of variation (CV) values further indicated the potential of iRFP713 for NIR-II fluorescence imaging.

### NIR-II in vivo imaging of iRFP713
To detect whether iRFP713 could be readily used in the analysis of mammalian tissue, we infected wild-type C57BL/6 mice with *iRFP713*-carrying adeno-associated virus (AAV-iRFP713) by intravenous injection. Ten days after infection, we performed whole-body NIR-II imaging of the mice (Supplementary Fig. 3a) and were able to observe clear and high contrast in vivo fluorescence signaling within the middle area of the abdomen (Supplementary Fig. 3b), indicating that the virus successfully delivered the iRFP713 gene into the body. We further carried out an ex vivo experiment and found that while many organs exhibited iRFP713 fluorescence, this fluorescence was brightest in the liver (Supplementary Fig. 3b). These data suggest the utility of iRFP713 for noninvasive NIR-II imaging of a living body.

It is well known that systemic delivery of iRFP713 using an AAV will result in a significant accumulation of the probe in the liver. While this may be desirable for some applications, for others, it may be preferable to control iRFP713 expression in target tissues over specific time periods. Therefore, we adopted CRISPR/Cas9-based genome editing to knock the iRFP713 gene into the endogenous *Rosa26* locus in mice (iRFP713^flox/flox). Specifically, we placed a lox-stop-lox (LSL) element in front of *iRFP713* following the CAG promoter (Supplementary Fig. 4a), preventing the expression of iRFP713 unless a Cre recombinase is introduced to remove the stop cassette. Genotyping analysis certified the successful construction of the desired transgenic mice (Supplementary Fig. 4b). Upon crossing iRFP713^flox/flox mice with different tissue-specific Cre lines, the stop element, flanked by two lox sites, could be deleted or inverted through Cre-mediated recombination, leading to iRFP713 activation only in Cre-expressing tissues. Therefore, using this technique, we were able to establish a tissue-specific iRFP713-expressing transgenic mouse (Fig. 2a). To evaluate its imaging capacity, we generated a whole-body iRFP713-expressing mouse (iRFP713^flox/flox; EIIa-Cre) by crossing the iRFP713^flox/flox mouse with the EIIa-Cre line[30]. After confirming that there were no apparent side effects or health concerns in the mouse, we examined the fluorescence characteristics of the desired offspring. The entire body of the iRFP713^flox/flox; EIIa-Cre mice exhibited bright NIR-II fluorescence, in contrast to their control littermates (iRFP713^flox/flox) (Fig. 2b and Supplementary Movie 1).

We further detected the fluorescence of the iRFP713-expressing mice at 1, 3, and 8 weeks and even 15 months after birth. The results showed that the iRFP713 fluorescent signals could be clearly detected through NIR-II imaging across all these time points (Fig. 2b, c). In addition, we were able to distinguish the boundaries of certain organs, such as the liver, and observe the details of inner tissues when viewed from different angles (Fig. 2c). To assess the distribution of iRFP713, the mice were sacrificed, and the major organs, including the heart,

liver, pancreas, lung, kidney, and stomach, were harvested. No significant differences were observed between control and iRFP713-expressing mice in terms of organ morphology. We then examined the iRFP713 gene expression level by RT–qPCR (Supplementary Fig. 5a) and the protein level by normalizing the fluorescence intensity (Supplementary Fig. 5b). Moreover, we performed ex vivo NIR-II fluorescence imaging on isolated organs (Fig. 2d). The fluorescence intensities of different organs varied, from minimal in the kidney and stomach to higher levels in the heart, lung, and pancreas to especially high in the liver (Fig. 2d, e and Supplementary Fig. 5c). Together, these data indicated that iRFP713-based NIR-II fluorescence imaging could serve as a platform for noninvasive tissue observation.

## Real-time tracing of liver regeneration in vivo

To test whether our imaging system holds potential for the long-term observation of intravital biological processes, we used liver regeneration as a monitoring model. To prevent iRFP713 fluorescence interference from nontarget tissues other than the liver, such as that demonstrated in whole-body iRFP713-expressing mice, we considered the development of a hepatocyte-specific iRFP713-expressing animal more suitable. For this purpose, we crossed iRFP713 flox/flox mice with an Alb-Cre line, in which Cre recombinase is under the control of the Alb promoter, to allow Cre expression exclusively in hepatocytes (Fig. 3a). As expected, the livers of both young (8 weeks) and aged (15 months) iRFP713 flox/flox; Alb-Cre mice could be clearly visualized by NIR-II fluor-

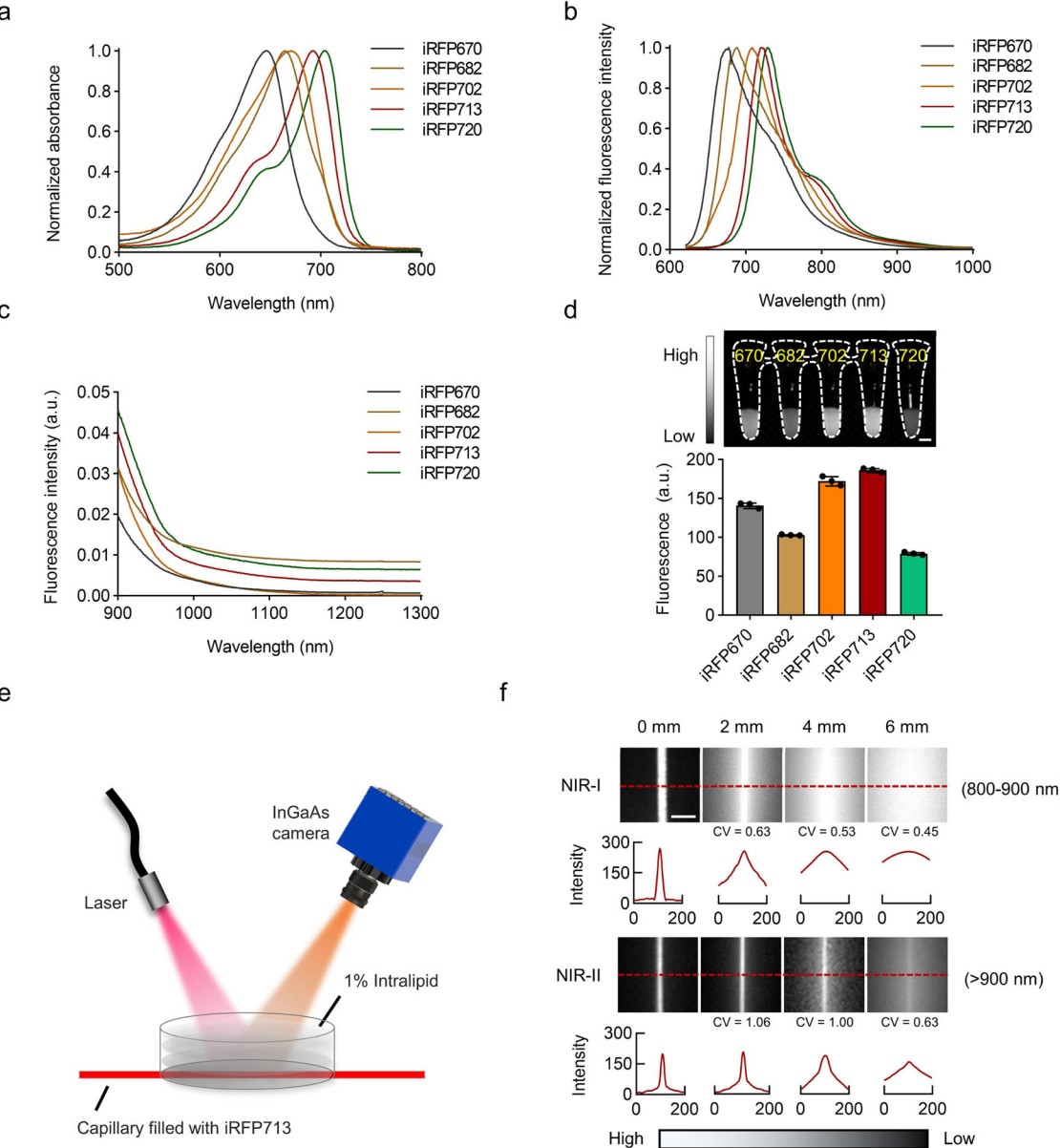

**Fig. 1 | Characterization of iRFPs. a, b** Normalized absorption spectra (**a**) and fluorescence emission spectra (**b**) of five purified iRFPs. (**c**) Emission spectra of iRFPs in the range of 900-1300 nm. **d** NIR-II fluorescence imaging of five purified iRFPs (0.3 mg mL$^{-1}$, 690 nm excitation (Ex) at ~60 mW cm$^{-2}$ and 900 nm LP emission (Em), 15 ms). Relative fluorescence intensity quantification is provided (lower panel). Data presented as mean values ± SEM, calculated from three independent experiments performed in triplicate. **e, f** Schematic diagram (**e**) and NIR

fluorescence images (**f**) of a capillary tube containing iRFP713 immersed at various depths in 1% intralipid, recorded under different LP filters (1 mg mL$^{-1}$, 690 nm Ex and 800-900 nm/>900 nm Em, respectively). Exposure times: (NIR-I) 0 mm and 2 mm: 10 ms; 4 mm: 80 ms; 6 mm: 100 ms. (NIR-II) 0 mm and 4 mm: 10 ms; 2 mm: 5 ms; 6 mm: 100 ms. Corresponding cross-sectional fluorescence intensity profiles along the indicated red-dashed lines were evaluated for each channel at each depth. Scale bars: 5 mm (**d, f**).

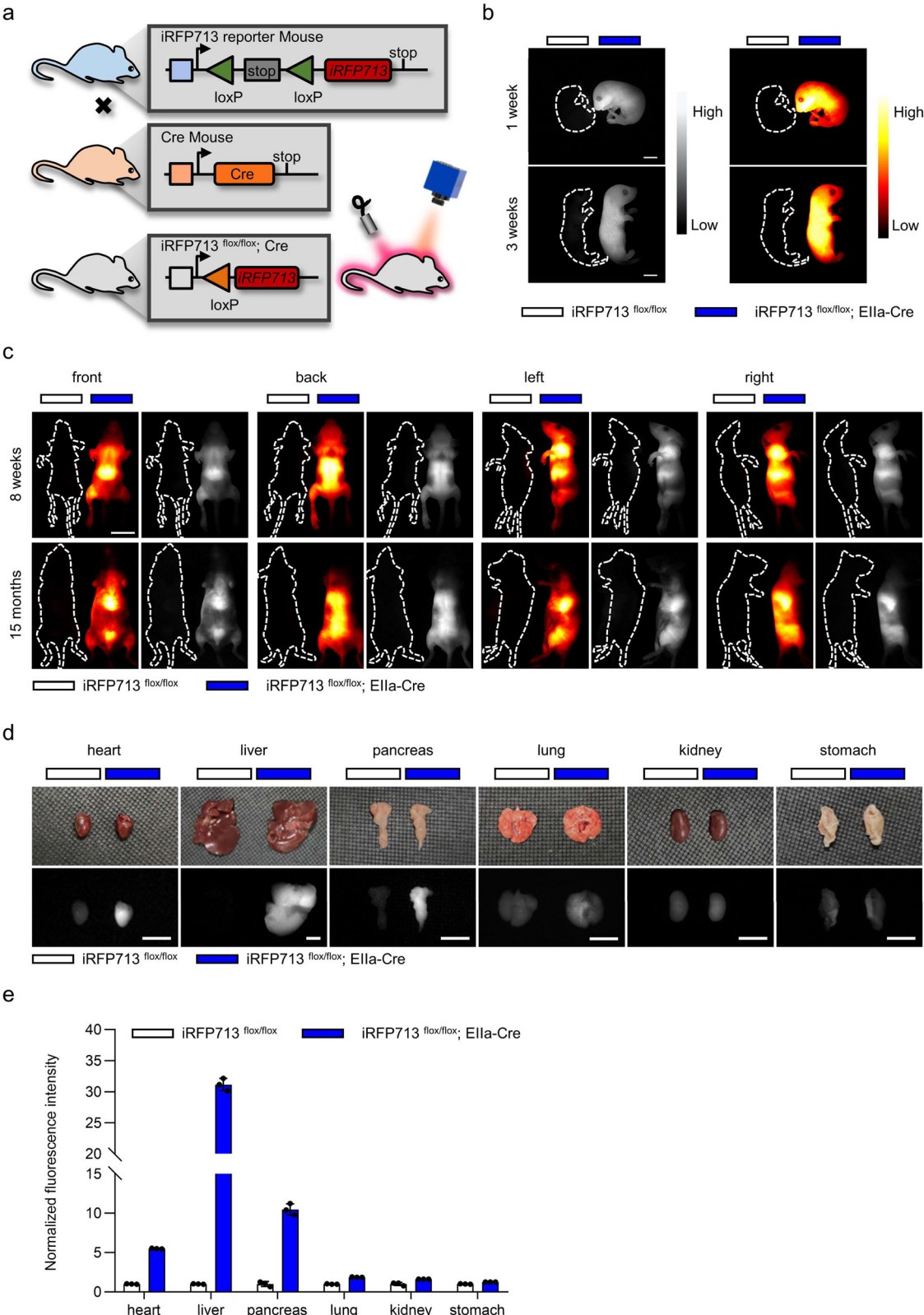

**Fig. 2 | Generation and NIR-II fluorescence imaging of inducible iRFP713-expressing mice. a** A brief illustration of the process of generating inducible iRFP713-expressing mice. **b, c** NIR-II fluorescence imaging of whole-body iRFP713-expressing mice (iRFP713^flox/flox; EIIa-Cre) and control littermate mice (iRFP713^flox/flox) at different times (1 and 3 weeks) after birth (**b**) and on different views (from the front, back, left, and right sides) (8 weeks and 15 months) (**c**). **d** Ex vivo NIR-II fluorescence images of major organs isolated from both mice provided in **c**. **e** Normalized fluorescence intensities of organs shown in **d**. Data presented as mean values ± SEM, $n = 3$ mice for each group. Images were acquired with 690 nm excitation and 900 nm LP emission filters. Exposure times: 35 ms (**b, c**) and 20 ms (**d**). Scale bars: 1 cm (**b**), 2 cm (**c**) and 5 mm (**d**).

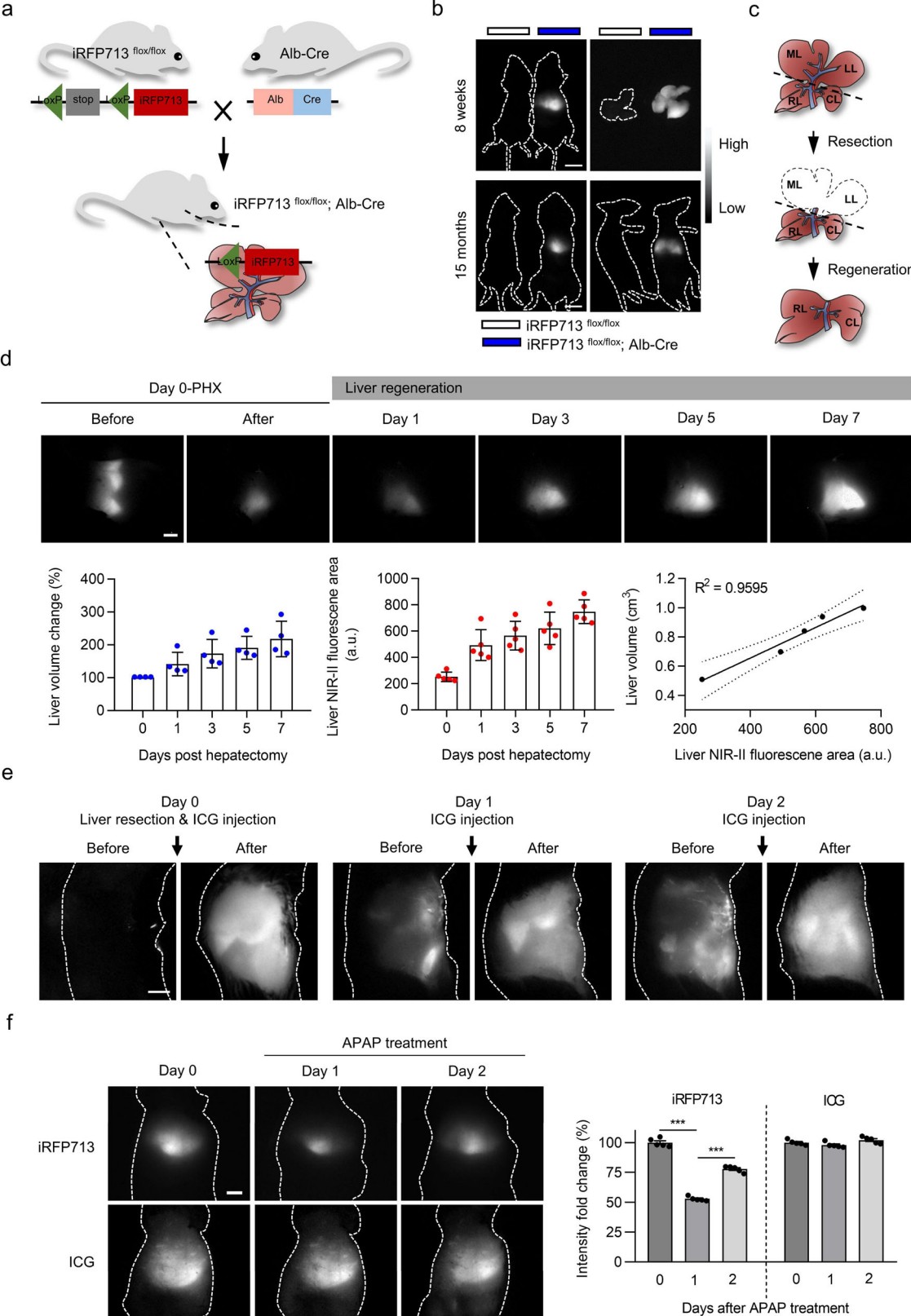

escence imaging, whereas the control littermates did not exhibit any fluorescence (Fig. 3b). Additionally, ex vivo imaging showed that the fluorescence was specifically displayed in the liver and was lacking in the other organs (Supplementary Fig. 6). These data suggest that the hepatocyte-specific iRFP713-expressing mouse could serve as a specific NIR-II imaging platform for in vivo liver observation.

Next, we established a partial hepatectomy (PHX)[31] model in iRFP713flox/flox; Alb-Cre mice to monitor the process of liver regeneration. In this model, approximately two-thirds of the liver is surgically removed, and the liver then restores itself to its original mass within 7–10 days by the proliferation of hepatocytes[32] (Fig. 3c and Supplementary Fig. 7). As shown in Fig. 3d, NIR-II fluorescence imaging was

**Fig. 3 | Noninvasive real-time monitoring of liver regeneration via NIR-II fluorescence imaging based on hepatocyte-specific expression of iRFP713.**
**a** Schematic diagram for the generation of hepatocyte-specific iRFP713-expressing mice (iRFP713^flox/flox; Alb-Cre). **b** NIR-II fluorescence images of the whole body of both mice (8 weeks and 15 months) and their isolated livers. The control mouse/liver is shown on the left. **c** Schematic illustration of PHX-induced liver regeneration. ML: middle lobe; LL: left lobe; RL: right lobe; CL: caudate lobe. **d** Upper section: in vivo NIR-II fluorescence imaging of the liver for iRFP713^flox/flox; Alb-Cre mice at various time points before and after PHX. Lower section: liver volumes as measured by enhanced μCT scanning (left panel) ($n = 4$ mice), liver fluorescence area calculated from the NIR-II images (middle panel) ($n = 5$ mice), and the correlation between liver volume (cm³) and fluorescence area (a.u.) (right panel). **e** ICG-assisted NIR-II fluorescence imaging of the liver in C57BL/6 mice with PHX at different time points. **f** NIR-II fluorescence imaging assisted by iRFP713 (in iRFP713^flox/flox; Alb-Cre mice) or ICG (in C57BL/6 mice) before and after APAP (250 mg/kg) treatment ($n = 5$ mice for each group). Images were taken with 690 nm Ex/900 nm LP Em and 793 nm Ex/900 nm LP Em for iRFP713 and ICG, respectively. The fluorescence intensity quantification for the liver is shown on the right (***$p < 0.001$). Data presented as mean values ± SEM. Statistical analysis was performed by one-way ANOVA. Exposure times: 15 ms (**b**, **e**, **f**) and 10 ms (**d**). Scale bars: 1 cm (**b**) and 5 mm (**d–f**).

able to clearly decipher a normal 'crown-like' shape containing three parts (middle, left, and right lobes) in a pre-PHX mouse when viewed from the right and to visualize the remnant liver after the middle lobe and left lobe were removed. The area and intensity of fluorescence then gradually increased from day 1 to day 7 with the corresponding recovery and regeneration of the liver (Fig. 3d). Quantitative analysis of imaging revealed a slight increase in hepatic fluorescence area 1 day after PHX, reaching a 2-3-fold increase by days 5–7. Furthermore, significant increases in liver fluorescence intensity were observed during regeneration due to enhanced iRFP713 expression accompanying hepatocyte proliferation. Notably, our results revealed that increased iRFP713 expression did not affect liver proliferation, as demonstrated by unaltered cyclin D1 and Ki-67 expression (Supplementary Fig. 8a–e), but was positively correlated with hepatocyte proliferation after PHX (Supplementary Fig. 8f–g).

To provide a reference standard, we also quantitatively measured the remnant liver volume via contrast-enhanced μCT, which has been commonly used in clinical settings for hepatic diagnosis[33]. The μCT analysis revealed that liver volume was significantly increased after resection, up to 191.0% ± 35.0% and 218.3% ± 53.8% at day 5 and day 7, respectively (Fig. 3d). We then estimated the correlation of liver measurement data from μCT compared with iRFP713-assisted NIR-II fluorescence imaging. The iRFP713 fluorescence area data correlated well with the volumes measured by μCT ($R^2 = 0.9595$, $p < 0.01$) (Fig. 3d), indicating that our system is comparable to the classic μCT in visualizing the mouse liver.

Indocyanine green (ICG) is a clinically approved safe fluorescent dye used in hepatic studies and NIR fluorescence imaging[34]. To compare the performance of iRFP713 and of this dye, we conducted NIR-II imaging using ICG in the PHX model. Unlike iRFP713, we had to separately administer ICG through the tail vein prior to each observation. ICG-assisted imaging did not provide clear liver-to-other tissue contrast, as the dye was rapidly metabolized and unable to specifically accumulate in the liver (Fig. 3e). Furthermore, we observed that ICG eventually entered the intestine through the biliary tract after several injections, leading to persistent fluorescence in the intestine, as previously reported[35] (Fig. 3e and Supplementary Fig. 9a). Liver function analysis also revealed that ICG administration exhibited hepatotoxicity with elevated alanine transaminase (ALT), aspartate transaminase (AST), alkaline phosphatase (ALP) and total bilirubin (TBIL) levels at day 1 compared to the controls (Supplementary Fig. 9b). More importantly, although ICG did not alter hepatic proliferation (Supplementary Fig. 10a–c), it triggered severe hepatocyte apoptosis, as indicated by the increased number of TUNEL-positive cells (Supplementary Fig. 10d). These results demonstrated that iRFP713-assisted NIR-II fluorescence imaging of the liver has great advantages over the traditional ICG method.

We also adopted a classic chemical liver injury model, acetaminophen (APAP)-driven liver regeneration[36] (Supplementary Fig. 11), to evaluate the visualization capabilities of iRFP713 and ICG. iRFP713-assisted imaging revealed that the hepatic fluorescence intensity declined by 62.8% at day 1 (Supplementary Fig. 12a and Fig. 3f) due to the APAP-induced death of hepatocytes (Supplementary Fig. 12b–d)

but then increased by 17.4% and 35.1% at days 2 and 3, respectively (Fig. 3f). This 'first-fall-then-rise' phenomenon is consistent with the pathological process of acute liver injury, which displays distinct injury and resolution phases[37]. However, no significant changes were observed in ICG-based imaging (Fig. 3f).

Hence, we conclude that iRFP713-assisted NIR-II fluorescence imaging is a superior approach for liver visualization.

## Noninvasive long-term visualization of the pancreas

Deep tissue imaging provides a particular challenge. As one of the deeply located organs, the pancreas has both endocrine and exocrine functions and is clearly related to common disease processes such as diabetes and pancreatitis[38, 39]. To test how our system works in this deeply located tissue, we crossed the iRFP713^flox/flox with the Pdx1-Cre strain to generate pancreas-specific iRFP713-expressing mice (iRFP713^flox/flox; Pdx1-Cre) (Fig. 4a). Clear fluorescence in the pancreas was evident under NIR-II imaging regardless of whether the mouse was young (8 weeks) or aged (15 months), while their control littermates lacked any obvious NIR-II fluorescence signals (Fig. 4b). We then confirmed pancreas-specific iRFP713 fluorescence in this transgenic mouse by detecting fluorescent signals from the internal pancreas during laparotomy (Supplementary Movie 2) in the fluid-filled pancreas and the isolated pancreas (Fig. 4c).

To verify the pancreas imaging capacity under pathological conditions, we first used streptozotocin (STZ) to produce a diabetic model (Supplementary Fig. 13a, b). Here, iRFP713^flox/flox; Pdx1-Cre mice were administered multiple low doses of STZ to induce pancreatic β cell exhaustion and death. We then performed NIR-II fluorescence imaging at different time points. As shown in Fig. 4d, the fluorescence signals of the pancreas were slightly decreased on days 1, 3, 5, and 7 after STZ injection, but the differences were not statistically significant (Fig. 4d and Supplementary Fig. 13c). We considered this to be mainly because of the low percentage of iRFP713-expressing β cells in the whole pancreas. To meet this challenge, we administered cerulein to mice to induce acute pancreatitis (Fig. 4e). In this model, pancreatic acinar cells, the main iRFP713-expressing cells, are the target of cerulein. Imaging results showed that iRFP713 fluorescence dramatically declined after 1 day of intraperitoneal injection of cerulein, followed by a significant increase in fluorescence as the pancreas recovered from acute pancreatitis (Fig. 4f). These data indicated that NIR-II imaging of pancreas-specific iRFP713 could be utilized to monitor the pancreas under both physiological and pathological conditions.

## Discussion

NIR-II fluorescence imaging has been widely used in both basic research and clinical practice[40–42]. However, its use is plagued by intrinsic defects associated with the synthetic nature of the probes, including their nonrenewable feature, proliferation-linked signal dissipation, rapid breakdown and elimination from the body, low biocompatibility, and the presence of false-positive readings caused by off-target tissue uptake. Such issues have restricted its broader uses in various biomedical scenarios, particularly for long-term application. In this study, we found that iRFPs, a kind of biologically rather than

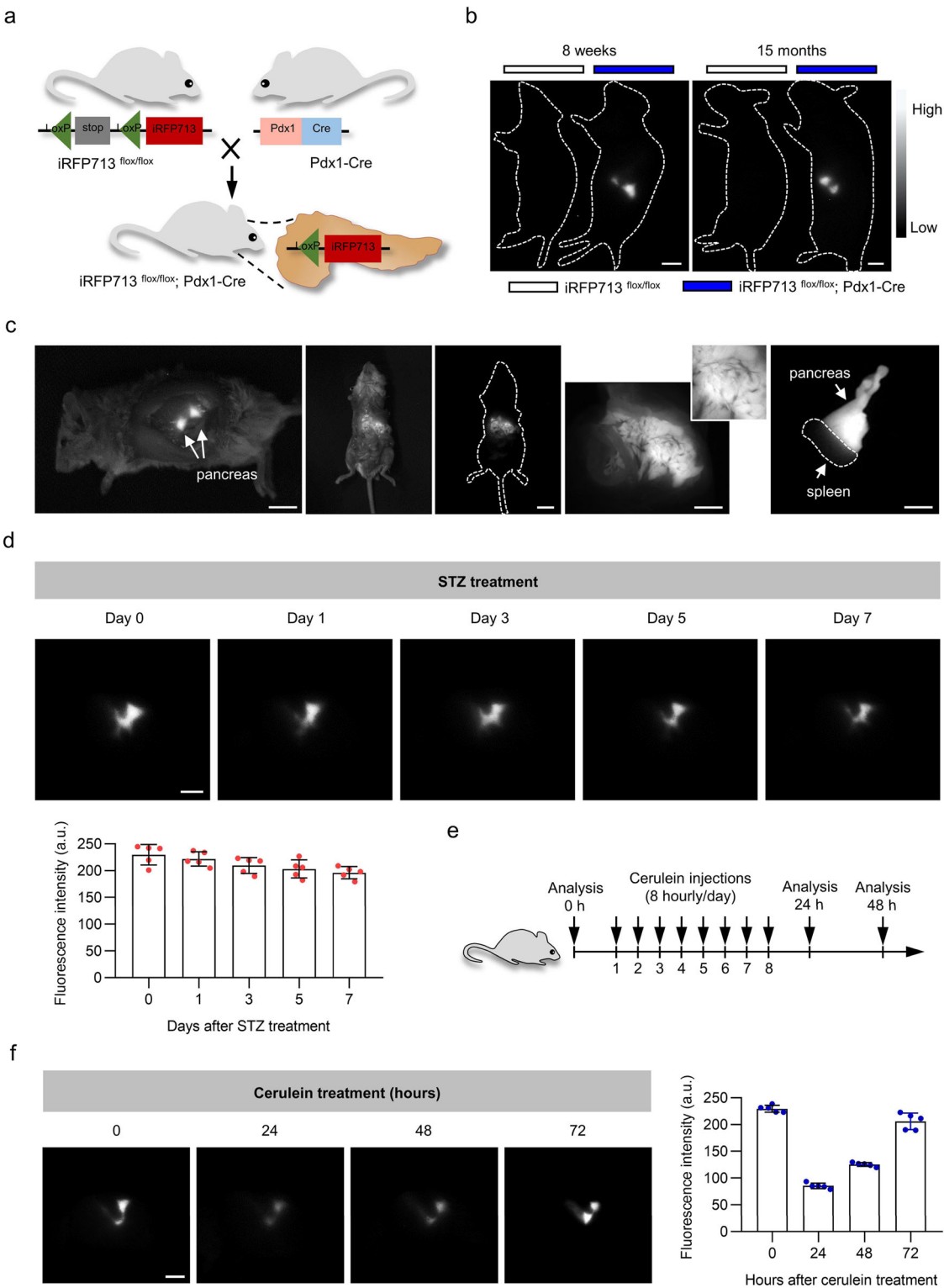

Fig. 4 | iRFP713-assisted noninvasive in vivo imaging of the pancreas under physiological and pathological conditions. a Schematic illustration of the generation of pancreas-specific expression of iRFP713 mice (iRFP713[flox/flox]; Pdx1-Cre). b Whole-body imaging of iRFP713[flox/flox]; Pdx1-Cre mice and control mice (8 weeks and 15 months). c NIR-II fluorescence imaging of the pancreas with laparotomy (left panel), the fluid-filled pancreas (middle three panels. Insert: the magnified image of the fluid-filled pancreas), and the isolated pancreas (right panel) of iRFP713[flox/flox]; Pdx1-Cre mice. d NIR-II fluorescence imaging of the pancreas for iRFP713[flox/flox]; Pdx1-Cre mice at various time points (days 0, 1, 3, 5, and 7) after STZ injections. The

fluorescence intensity quantification for the pancreas is shown in the lower panel (n = 5 mice). e Brief description of the cerulein-induced pancreatitis experiment. Mice were administered 8 hourly doses of cerulein to induce acute pancreatitis. f NIR-II fluorescence imaging of the pancreas for iRFP713[flox/flox]; Pdx1-Cre mice at various time points (0, 24, 48, 72 hours) after cerulein injections. Fluorescence intensity quantifications are shown to the right (n = 5 mice). Data presented as mean values ± SEM. Exposure times: 20 ms (b, d, f) and 10 ms (c). Scale bars: 5 mm (b, d, f) and 1 cm (c).

chemically synthesized probe that can avoid many of the mentioned deficiencies, had NIR-II spectral signatures and could be developed as a long-term NIR-II fluorescence imaging platform.

As biocompatible proteins, iRFPs not only are renewable because the probe gene can be genetically encoded into the genome to achieve a signal amplification of fluorescence upon proliferation of the target cells but also make tissue-specific observation a growing reality. As demonstrated in this study, the liver and pancreas could be specifically genetically coded to express the protein probe through a Cre-LoxP approach. In theory, the labeling is not time-limited, and in our case, the transgenic mice had stable iRFP713 gene and protein expression for at least 15 months (Supplementary Fig. 14). Since various tissue-specific Cre lines have already been generated, with others soon to be completed, we believe that every organ in the body could be represented by this technology. The mouse heart, for example, displayed bright fluorescence in our results, and we were able to apply iRFP713 successfully for cardiac imaging with the help of a cardiac myocyte-specific Cre driver line[43]. Moreover, the temporally inducible Cre-LoxP system can turn on iRFP713 expression in a time-specific manner with the use of an exogenous inducer. Thus, we could induce iRFP713 expression to visualize the morphology of particular tissues at certain time points (for example, during embryonic development or tumorigenesis). With these advantages, iRFP713-based NIR-II fluorescence imaging enables the monitoring of specific cells or tissues at precise positions and times.

To confirm the superiority of iRFP713-assisted NIR-II fluorescence bioimaging, the iXon Ultra 897 camera (EMCCD with ultrahigh sensitivity) and the GA1280 camera (vis-NIR camera with high quantum efficiency within 800–900 nm) were selected for NIR-I detection (800–900 nm), while a typical InGaAs camera (SW640) was utilized for NIR-II detection (beyond 900 nm). The in vitro intralipid phantom imaging and in vivo imaging of transgenic mice (Supplementary Fig. 15 and Supplementary Table 1) using the above three cameras showed that images recorded in the NIR-II window had the best imaging quality, whether the EMCCD or vis-NIR camera was used for NIR-I detection. Thus, we believe that the excellent imaging performance in the NIR-II window should be credited to the window itself.

Nevertheless, the emission of iRFP713 beyond 900 nm remains relatively weak due to the low tailing proportion. To address this problem, it is necessary to either develop FPs with longer excitation and emission spectral responses, perhaps by using structure-guided multisite mutagenesis[44], or incorporate an advanced InGaAs camera with higher sensitivity for long-wavelength detection. Either or both of these approaches may be soon within reach. Moreover, certain organs showed lower fluorescence intensities in our study, such as the kidney and stomach. It is well known that many factors affect the fluorescence intensity of tissue, including its depth in the body, the number of iRFP-expressing cells, and the strength of the tissue-specific Cre promoter. To improve the iRFP713 expression level, one solution would be to select a strong promoter for this particular tissue, such as stomach-specific Cre: Atp4b-Cre[45] and kidney-specific Cre: Ksp 1.3-Cre[46], to promote iRFP713 expression.

Overall, by employing protein iRFP713 as a probe, we have developed a biocompatible and renewable in vivo NIR-II monitoring approach with deep penetration and high imaging contrast. This platform is not only suitable for single-tissue observation when prepared for tissue-specific expression but also able to achieve prolonged visualization of any organ. We believe that this platform could serve as a powerful tool for the long-term monitoring of many intravital biological processes.

## Methods

### Ethical statement
All animal studies were performed in compliance with the guide for the care and use of laboratory animals, and the protocol was approved by the Medical Experimental Animal Care Commission of Zhejiang University (#ZJU20200311).

### Mice
C57BL/6 mice aged 6–8 weeks were purchased from the Shanghai SLAC Laboratory Animal. iRFP713 transgenic mice were generated by Cyagen Biosciences using CRISPR/Cas9 on a C57BL/6 background. Pdx1-Cre, Alb-Cre, and EIIa-Cre mice were purchased from Cyagen Biosciences. Mice were maintained and bred in specific pathogen-free conditions with a 12 h light and 12 h dark cycle, 25 °C room temperature and 50.0 ± 5.0% humidity at Laboratory Animal Center of Zhejiang University. The food and water were provided *ad libitum*.

### iRFPs purification and characterization
Briefly, iRFP670, iRFP682, iRFP702, iRFP713, and iRFP720 were expressed with an N-terminal 6xHis tag in a modified pET28a expression vector (Invitrogen) containing the heme oxygenase-1 encoding gene from cyanobacteria. Proteins were purified using the Ni-NTA purification system (QIAGEN) according to the manufacturer's protocol. The protein concentration was measured using a BCA protein assay (Thermo Fisher Scientific). The absorption spectra of iRFPs were obtained from 500–800 nm with a Shimadzu UV-2550 UV–vis–NIR scanning spectrophotometer. Fluorescence spectra of iRFPs were measured using a home-built system based on a spectrometer (FLS980, Edinburgh Instruments).

### Mammalian plasmid construction and stable cell line generation
HEK293T (GNHu17), HEK293 (GNHu43), Huh7 (SCSP-526), and HT29 (SCSP-5032) cell lines were purchased from Shanghai Cell Bank of the Chinese Academy of Sciences. The iRFP713 gene was amplified by PCR and inserted into a pCDH-CMV-MCS-EF1-Puro vector (System Biosciences) at the HindIII/NotI site to generate a pCDH-CMV-iRFP713-EF1-Puro plasmid. To generate stable cell lines, lentivirus particles were produced by transient cotransfection of HEK293T cells with the pCDH-CMV-iRFP713-EF1-Puro plasmid and helper vectors psPAX2 and pMD2. G (Invitrogen) using Lipofectamine 2000 reagent (QIAGEN) according to the manufacturer's protocol. The target cell lines, HEK293, Huh7, and HT29 cells, were infected with the virus-containing supernatant for 48 h and selected with puromycin antibiotic for 2 weeks. The iRFP713 mRNA expression level was examined by RT–PCR.

### NIR-I fluorescence imaging
Intralipid phantom imaging in the NIR-I window was carried out with a lab-built NIR-I fluorescence imaging setup equipped with a 690 nm laser. The beam was coupled to a collimator and expanded by a lens, providing uniform irradiation on the imaging plane. Images were captured using a silicon camera (512 pixels × 512 pixels, iXon Ultra 897, Andor) equipped with a prime lens (focal length: 50 mm, antireflection coating at 800-2000 nm, Edmund Optics), which was fitted with two 800 nm longpass filters (Thorlabs) and a 900 nm shortpass filter (Thorlabs) to extract the NIR-I fluorescence signal.

### In vitro NIR-II fluorescence imaging
For intralipid phantom imaging in the NIR-II window, a 690 nm laser beam was coupled to a collimator and expanded by a lens providing uniform illumination of the field. An InGaAs camera (640 pixels × 512 pixels, 900–1700 nm sensitive, TEKWIN SYSTEM) fitted with a 900 nm longpass filter (Thorlabs) was used to detect the NIR-II fluorescence signals.

For high-magnification NIR-II fluorescence microscopic imaging of the stable cell lines, we used the system described in our previous study[15]. In brief, a 690 nm laser, which was reflected by a 900 nm longpass dichroic mirror (Thorlabs), passes through an infrared anti-reflection water-immersed objective lens (XLPLN25XWMP2, 25×, numerical aperture (NA) = 1.05, Olympus) and then illuminated the

cells. The NIR-II fluorescence signals were detected using an InGaAs camera with a 900 nm longpass filter (Thorlabs).

## Generation and characterization of iRFP713 transgenic mice

To generate iRFP713 transgenic mice, the donor vector containing the "CAG promoter-loxP-3xpolyA-loxP-Kozak-iRFP713-polyA" cassette was constructed and subsequently inserted into intron 1 of the Rosa26 gene at position +113025988 of mouse chromosome 6. To generate targeted conditional knockin offspring, the guide RNA for mouse Rosa26 (matching forward strand of gene; TGGGCGA-GAAATGTGTCCTG), Cas9 mRNA, and the target donor vector were coinjected into fertilized mouse eggs. F0 founder mice were identified by PCR followed by sequence analysis and were bred with wild-type mice to test germline transmission and F1 animal generation.

For mouse genotyping, genomic DNA was extracted from the mouse tail by alkaline lysis, and then PCR was performed to genotype the mice. For tissue expression of iRFP713, mRNA and protein were extracted from different tissues of the mice. After protein concentration quantitation, tissue protein samples were imaged by a lab-built in vitro NIR-II fluorescence imaging system. The accurate quantification of iRFP713 gene expression was conducted using RT-qPCR. The sequences of all primers are provided in Supplementary Table 2.

## Liver regeneration model

To establish a liver regeneration model, partial hepatectomy (PHX) was performed in 8-week-old male iRFP713[flox/flox]; Alb-Cre mice under sterile conditions, as previously reported[31]. Briefly, the mice were anesthetized with isoflurane and placed on a heating pad to maintain body temperature. After midline laparotomy, the left and middle lobes of the liver were ligated and surgically resected at a level 3 mm proximal to the ligation. After resection, NIR-II fluorescence in vivo imaging was conducted on days 1, 3, 5, and 7.

To establish a liver regeneration model induced by acetaminophen (APAP) (Sigma Aldrich), 2- to 3-month-old male iRFP713[flox/flox]; Alb-Cre mice were starved for 12 h with free access to water and injected intraperitoneally with 250 mg/kg APAP dissolved in warm 0.9% saline at a concentration of 15 mg/mL. After IP injection of APAP, the mice were returned to normal food. NIR-II fluorescence imaging of mice was performed at 1 and 2 days after APAP treatment.

## Proliferation and apoptosis assays

Tissue samples from mice of the corresponding genotypes were isolated, weighed and divided into aliquots for RT-qPCR, western blot, hematoxylin and eosin (HE) staining, and immunohistochemical (IHC) staining. Cell proliferation was evaluated by Ki-67 IHC staining according to the standard procedure. A TUNEL assay was performed to investigate cell apoptosis according to the manufacturer's manual (Elabscience). The primary anti-iRFP713 antibody was produced by HuaAn Biotechnology, and the specificity was verified by western blotting. Other antibodies used in this study are available in Supplementary Table 3.

## Streptozotocin-induced diabetic model

Eight-week-old male iRFP713[flox/flox]; Pdx1-Cre mice were repeatedly injected intraperitoneally with 50 mg/kg streptozotocin (STZ) (Sigma Aldrich) to induce a diabetes model, as previously described[47]. Briefly, immediately prior to injection, STZ was reconstituted in cold, pH 4.5, 25 mM sodium citrate buffer (Sigma Aldrich). Injections were administered intraperitoneally within 20 min of preparation. Mice were treated with five consecutive daily injections, and NIR-II fluorescence in vivo imaging was performed at 1, 3, 5, and 7 days after STZ administration. Blood glucose levels were measured by drawing blood from the tail vein at the indicated time points. Pancreatic tissues were harvested from mice, and tissue sections were routinely stained with hematoxylin and eosin.

## Cerulein-induced pancreatitis model

To establish an acute pancreatitis model, 8-week-old male iRFP713[flox/flox]; Pdx1-Cre mice were treated with cerulein (Sigma Aldrich). Briefly, before the experiment, the mice were fasted overnight and allowed water ad libitum. Mice were then injected intraperitoneally with 50 μg/kg cerulein dissolved in 0.9% saline in a volume of 100 μL. Cerulein injections were undertaken in hourly intervals for up to 7 injections. After injections, the mice were returned to the normal diet. NIR-II fluorescence in vivo imaging of mice was conducted before the first injection and 24, 48, and 72 hours afterward.

## In vivo NIR-II fluorescence imaging

The macroimaging system was described in our previous study[5]. Briefly, a 900 nm longpass filter was mounted in front of the InGaAs camera. By using this imaging system, in vivo NIR-II fluorescence imaging of different mouse models was conducted, i.e., the mice were irradiated under a 690 nm laser for iRFP713 NIR-II fluorescence imaging. For indocyanine green (ICG)-assisted NIR-II fluorescence imaging, 1 mg/kg ICG was intravenously injected through the tail vein and then studied using excitation at 793 nm.

## Image processing

The fluorescence images were collected from the customized software of TEKWIN SYSTEM (SW640). Quantitative analysis of the fluorescent images was performed using ImageJ software (Version 1.6.0, National Institutes of Health) based on the measurement of mean signal intensity in the manually selected regions of interest. All images were processed using the same settings within a test for both the control and experimental groups. For NIR-II imaging of the liver, the grayscale image sequence was taken and binarized by the same threshold in image preprocessing. According to the binarized image, image segmentation was performed with light as the signal area and dark as the background area. The sum of the value on each pixel of the bright area was considered to be the total fluorescence intensity, as was the intensity of the background signal. Then, the signal-to-background ratio (SBR) was calculated.

## Statistics and Reproducibility

Data were analyzed using OriginPro 2018 (64 bit) and GraphPad Prism 8 (GraphPad Software). Two-tailed Student's $t$ test (where two groups of data were compared) and one-way ANOVA (where more than two groups of data were compared) were used to evaluate the significance of the differences. $P$ values <0.05 was considered statistically significant. Statistical significance was defined as $*P < 0.05$, $**P < 0.01$ and $***P < 0.001$.

## Reporting summary

Further information on research design is available in the Nature Research Reporting Summary linked to this article.

# Data availability

All the data generated in this study are provided in the Supplementary Information/Source Data file. There are no restrictions on data availability in the current work. Source data are provided with this paper.

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

## Acknowledgements

This work was supported by the "Medicine + X" Interdisciplinary Doctoral Program of Zhejiang University, National Natural Science Foundation of China (32101015, 81972612, 61975172 and 82001874), National Key R&D Program of China (2019YFC1708701), Fundamental Research Funds for the Central Universities (2020-KYY-511108-0007), and China Postdoctoral Science Foundation funded project (2021M692826 and BX20220260).

## Author contributions

J.S., J.Q. and Z.X. conceived and designed the project. M.C., Z.F., and X.F., performed the mouse experiments. J.S. conducted the cell experiments. M.C., Z.F., X.F., W.G., and T.W. performed the fluorescent imaging experiments and analyzed the data. M.C. and Z.F. wrote the original draft. J.S., J.Q., and Z.X. wrote and reviewed the final version of the text. J.S., J.Q., and Z.X. supervised the research.

## Competing interests

The authors declare no competing interests.
