## [Peer Review File · Nature Communications]

Reviewers' Comments:

Reviewer #1:

Remarks to the Author:

This manuscript focuses on the use of infrared fluorescent proteins (iRFPs) as in vitro and in vivo NIR-II probes. This was performed by using the emission tail of iRFPs at over 900 nm region. They knocked one of bright iRFP (iRFP713) into the mouse genome to generate a transgenic model. Additionally, this paper adopted two liver regeneration models and successfully tracked the process for a week. However, it is somewhat disappointing that there is insufficient data to support their design. There is no direct comparison of imaging quality by iRFP713 in NIR-II and its commonly used imaging region in their in vivo applications. It fails to demonstrate the purpose of performing NIR-II imaging by using an existing fluorescent protein. Thus, the major shortcoming of the manuscript is that it does not provide substantial evidence for the readership of the field, as the idea of using tail emission of near infrared dye has been reported by several groups including themselves. Therefore, my recommendation is that the paper is not suitable for publication in its current form, but upon a major revision it may become publishable.

1. As the authors acknowledged that the idea of tail emission for NIR-II bioimaging was previously reported. However, the authors had some overstatements, for instance: the tail emission is only introduced in their own work "To explore the application of iRFPs in the NIR-II region, we first characterized the spectral properties of this probe and found that its emission tail extended into the 900-1300 nm region and exhibited high level of brightness." The Ms. under review is not clear about this point relative to the previous reports, that may not justify the novelty of their contribution.
2. For Figure 1f, it's not reasonable to have such differences of penetration and scattering by performing NIR-I imaging using iXon Ultra 897 camera and NIR-II imaging using InGaAs camera. Indeed, the penetration depth and scattering would be improved by moving the imaging window from 800-900 nm to >900 nm region, but the improvement is generally limited, as a silicon camera with enough quantum efficiency at 800-900 nm could provide much better imaging quality compared with InGaAs camera. Authors need to carefully compare the imaging quality between these two regions to avoid an overestimation of the imaging quality in the >900 nm window.
3. The selling point of the Ms. is that iRFPs provide much improved imaging penetration depth and contrast in the NIR-II window over their traditional NIR window. The authors argue in favor of this assumption based on capillary tube with Intralipid studies. This qualitative reasoning is not enough to support the in vivo imaging. The authors should better compare all the in vivo imaging of iRFPs under the NIR-I/II regions using both silicon and InGaAs cameras simultaneously, further helpful in determining the advantage of fluorescent protein assisted NIR-II imaging.
4. I believe they need to do a set of important control studies. The authors might want to provide signal to background ratios (SBR) of current NIR-II imaging outcome, and compare the SBR with NIR-I data of the same in vivo models collected from silicon camera.
5. The molecular biology data is relatively insufficient and weak. Could you shed light on the effect of the long-term fluorescence stability of transgenic model? Due to the lack of appropriate evidences and data, the statement of fluorescence up to 15 months is not convincing.
6. The NIR-II quantum yield (QY) is very important to evaluate the imaging ability of a contrast agent. A fluorophore with very low QY would fail to provide significantly improved imaging quality even under the NIR-II imaging window. The longer exposure time would increase the background signal.
7. The text of the Ms. needs to be revised in order to correct for typos, or to avoid awkward phrasing.
8. The number of significant digits is not consistent in the manuscript, for instance, "191.0% ± 34.98% and 218.3% ± 53.8%" in Page 8.
9. Imaging conditions are missing in the Figure legend Figure 2. It should be mentioned, otherwise it is confusing to the readers.

Reviewer #2:

Remarks to the Author:

In this manuscript entitled "Long-term monitoring of intravital biological process using fluorescent

protein assisted NIR-II imaging”, the authors tested non-invasive imaging using NIR fluorescent protein iRFP-expressing mice with longer wavelength 900-1880 nm (NIR-II).

Non-invasive imaging of living mice using NIR fluorescence is thought to be very powerful methods but spread of the technique is not enough so far, thus the data of this manuscript are very useful for many researchers.

Therefore, this reviewer think that it is very important that experimental details should be described sufficiently and precisely in this manuscript.

The experimental strategies are well designed and overall the data are clearly presented but in several points the descriptions are insufficient or not suitable.

Major points

1) Because they used CRISPR/Cas9 system to establish Tg mice, they should show the sequence of the guide RNA to reveal clearly the target site on the mouse genome.

2) Because the intensity of NIR-II fluorescence is thought to be quite low, the exposure times should be shown and image adjustment method should be described in detail.

3) In Fig1 some important experimental descriptions are insufficient.

Fig1a: this reviewer can not understand the reason why the 3D structure of iRFP is needed for this manuscript.

Fig1c: Wavelength used for excitation should be shown.

The authors show the spectra in 600 - 1400 nm and detail in 600 - 800 is not clear. The data should be showed in 600 - 1000 nm

Fig1d: the intensity of iRFP670 seems to be too high because excitation rate is only 10%(Fig1b) in 690 nm and emission is very low(Fig1c). The authors should explain this contradiction.

Fig1f: this reviewer think that the data clearly show the advantage of NIR-II than NIR-I in light scattering. However, in upper data, intensity remains to be high even in deeper conditions(4 mm, 6 mm) but scattering becomes clearly higher. Usually the relationship between intensity and scattering is tradeoff and scattering becomes higher, intensity becomes lower like in lower data. The authors should explain this point.

The quantitative data of scattering (for example CV value) might be very useful for many researchers and if it is possible please show calculated data of the scattering.

4) Normal mouse food and milk from mother mouse usually show high intensity of NIR autofluorescence and some low NIR-fluorescence mouse foods are commercially available. Change of mouse food before taking photo is very important experimental technique to avoid the effects of autofluorescence.

In Fig 2 not only iRFP fluorescence but also milk autofluorescence seems to be overlapped.

In Fig3 and Fig4 no autofluorescence of food is detected. If the authors change mouse food before taking photo, they should make clear that point.

Minor point

The previous report establishing EIIa-cre mouse should be cited.

Reviewer #3:

Remarks to the Author:

General Comments: In this manuscript, the authors have described a novel iRFP713 fluorescent protein based in vivo imaging technology. They have shown that iRFP can be used to track change

in organ size (liver and pancreas) in models of regeneration. Overall, the studies are well conducted and interesting. However, significant details of the experiments and the actual cellular localization and its effects on the cells of the new iRFP713 protein are missing.

Detailed Comments:

1. Why is iRFP713 expression so much higher in liver than other tissues, especially given the cre used is not liver specific?
2. Which cells in the liver express iRFP713?
3. The experiments shown in Fig 3 on PHX are interesting. However, more information is needed to rule out possibility that iRFP expression changes kinetic of proliferation and pathways involved in regeneration. Determine cell proliferation using either PCNA or Ki-67, and mRNA of Cyclin D1 is important. A correlation of these markers with iRFP fluorescence can be a good additional measure.
4. To compare ICG and iRFP, parameters of proliferation and injury should be measured in both groups at the same time.
5. The data on acetaminophen toxicity model need lot more work. It is known that APAP overdose does not affect liver weight to any appreciable degree. The injury is mainly centrilobular (zone 3) in the liver and hepatocytes around the necrotic zone regenerate. It is not clear why iRFP declines after APAP administration. It may be due to the fact that albumin promoter doesn't work as well or may be due to overall cell death. This needs to be investigated. More experimental details such as the exact protocol used (fasted non-fasted, was the food returned after APAP treatment) and profiles of liver injury (did APAP actually produce injury or not) and recovery should be provided.

Point-by-point response to reviewers' comments

Reviewer #1 (Remarks to the Author):

This manuscript focuses on the use of infrared fluorescent proteins (iRFPs) as in vitro and in vivo NIR-II probes. This was performed by using the emission tail of iRFPs at over 900 nm region. They knocked one of bright iRFP (iRFP713) into the mouse genome to generate a transgenic model. Additionally, this paper adopted two liver regeneration models and successfully tracked the process for a week. However, it is somewhat disappointing that there is insufficient data to support their design. There is no direct comparison of imaging quality by iRFP713 in NIR-II and its commonly used imaging region in their in vivo applications. It fails to demonstrate the purpose of performing NIR-II imaging by using an existing fluorescent protein. Thus, the major shortcoming of the manuscript is that it does not provide substantial evidence for the readership of the field, as the idea of using tail emission of near infrared dye has been reported by several groups including themselves. Therefore, my recommendation is that the paper is not suitable for publication in its current form, but upon a major revision it may become publishable.

Response: We thank the Reviewer for positively evaluating our work and providing clear requirements to improve the manuscript. Following your instruction, we have performed iRFP713 *in vitro* and *in vivo* imaging at both NIR-I and II regions, and directly compared their imaging quality. It can be clearly seen that iRFP713-assisted NIR-II bioimaging is of better quality compared to the NIR-I one, enabling high-performance long-term monitoring of intravital biological processes. In order to let the readers understand our study more directly, we also re-write the manuscript to state that the previous reports using tail emission of near infrared dyes are all chemically synthesized ones, not proteins. Hope the Reviewer will now find that this revised manuscript has enough data to support its design and appreciate this is the first study to explore the NIR-II imaging capacity of a fluorescent protein. Please see below for more details.

1. *As the authors acknowledged that the idea of tail emission for NIR-II bioimaging was previously reported. However, the authors had some overstatements, for instance: the tail emission is only introduced in their own work “To explore the application of iRFPs in the NIR-II region, we first characterized the spectral properties of this probe and found that its emission tail extended into the 900-1300 nm region and exhibited high level of brightness.” The Ms. under review is not clear about this point relative to the previous reports, that may not justify the novelty of their contribution.*

Response: We feel sorry that our description was not clear enough to let the Reviewer understand our study rationale. Please allow me introduce one more time: the first paragraph of the Introduction intends to emphasize the importance of fluorescence imaging at NIR-II region, and raise the key challenge is to develop fluorescent probes with bright NIR-II emissions. The second paragraph of the Introduction names the currently used NIR-II fluorescent probes, lists their limitations (non-renewable in living systems and potentially toxic), and proposes renewable and biocompatible NIR fluorescent proteins (FPs) with characteristically longer emission wavelengths are in high demand. Therefore, we acknowledged the reported NIR FPs in the third paragraph, and clearly stated “only the NIR-I region has been exploited”. Based on these overviews, we moved on to say “To explore the application of iRFPs in the NIR-II region, ...” in the fourth paragraph, and then briefly introduce our work. To avoid misunderstanding, we have added “because they are chemically synthesized,” in the second paragraph of the INTRODUCTION (Page 3, line 14-15) to point out the nature of previously reported probes.

According to our knowledge, all the reported NIR-II fluorescent probes belong to chemical catalog, and so far no NIR-II fluorescent protein had been explored. Therefore, we believe that our work is the first one to take the advantage of NIR-II tailing emission of a fluorescent protein (iRFP) for bioimaging.

2. For Figure 1f, it's not reasonable to have such differences of penetration and scattering by performing NIR-I imaging using iXon Ultra 897 camera and NIR-II imaging using InGaAs camera. Indeed, the penetration depth and scattering would be improved by moving the imaging window from 800-900 nm to >900 nm region, but the improvement is generally limited, as a silicon camera with enough quantum efficiency at 800-900 nm could provide much better imaging quality compared with InGaAs camera. Authors need to carefully compare the imaging quality between these two regions to avoid an overestimation of the imaging quality in the >900 nm window.

Response: Again, we are sorry that we did not provide clear evidence to make the Reviewer believe our conclusion is reasonable. It is true that the improvements in scattering suppression and the attenuation length (one quantitative index of photon penetration) are generally limited in the NIR-II region. However, in the area detection-based bioimaging, the moderate absorption increase of emission light also improves the fluorescence imaging performance in the window beyond 900 nm (*Light: Science & Applications*, 2021, 10(1): 1-18; *Proceedings of the National Academy of Sciences*, 2018, 115(37): 9080-9085). **In fact, due to the light absorption around 980 nm, the scattered photons of emission light (acting as background) have longer optical paths than the ballistic photons (acting as signal), resulting in more attenuation in the simulated deep tissue, thus effectively restraining the imaging background and giving better imaging performance.**

The Reviewer might consider that the quantum efficiency (QE) within 800-900 nm of our silicon camera (i.e., iXon Ultra 897 camera, EMCCD) is not high enough, thus leading to an inaccurate estimation. Actually, the detection efficiency of a camera can be simply evaluated by the product of a fluorophore's emission intensity and the camera's QE:

- 1) For the fluorophore's emission intensity, we performed detailed fluorescence analyses within 800-900 nm and beyond 900 nm in our revised manuscript. As shown in modified Fig.1, the emission spectrum of iRFP713 demonstrated that it is a typical NIR-I fluorophore with much stronger emission in the NIR-I region (such as 800-900 nm in this work) than the NIR-II region (mainly 900-1000 nm in this work).
- 2) For the camera's QE, in the window of 800-900 nm, the QE of the silicon camera (yellow area in Supplementary Fig. 15a) is already **over 40%**. In contrast, the average QE in 900-1000 nm of the InGaAs camera is **only ~34%**, and its sensitivity displays a drastic change in this window (blue area in Supplementary Fig. 15b). Meanwhile, very little NIR-II emission of the iRFP713 locates beyond 1000 nm, and the relatively high QE (80%) of InGaAs camera beyond 1000 nm does not play its role. Thus, we think that the QE of our silicon camera is already high enough, compared with that of InGaAs camera.

Therefore, based on the above analyses, we conclude that although the intensity detection efficiency of 800-900 nm with the silicon camera has its supremacy, the imaging beyond 900 nm using the InGaAs camera could provide better quality due to the **moderate tissue absorption and suppressed imaging background.**

To further support our conclusion, we chose a wavelength-extended camera (i.e., GA1280, Tekwin, China, Spectral response: 400-1200 nm) whose QE within 800-900 nm exceeds 70% (red area in Supplementary Fig. 15c). The intralipid phantom imaging result clearly showed that its imaging performance was not as good as that of our silicon camera (Supplementary Fig. 15d), further supporting that the QE of a camera is not the only quality-determining factor. We have added these data to new Supplementary Figure 15, and discuss it in the DISCUSSION section of the revised manuscript (Page 11, line 30-34 and Page 12, line 1-5), as following:

“To confirm the superiority of iRFP713-assisted NIR-II fluorescence bioimaging, the iXon Ultra 897 camera (EMCCD with ultrahigh sensitivity) and the GA1280 camera (vis-NIR camera with high quantum efficiency within 800-900 nm) were selected for NIR-I detection (800-900 nm), while a typical InGaAs camera (SW640) was utilized for NIR-II detection (beyond 900 nm). The *in vitro* intralipid phantom imaging and *in vivo* imaging of transgenic mice (Supplementary Fig. 15 and Supplementary Table 1) using the above three cameras showed that images recorded in the NIR-II window had the best imaging quality, whether the EMCCD or vis-NIR camera was used for NIR-I detection. Thus, we believe that the excellent imaging performance in the NIR-II window should be credited to the window itself.”

Supplementary Figure 15a-b. The quantum efficiency curve of (a) iXon Ultra 897 camera (EMCCD) and (b) SW640 camera (InGaAs FPA).

Supplementary Figure 15c. The quantum efficiency curve of GA1280 camera (vis-NIR 2D-detector, black line).

Supplementary Figure 15d. NIR-I fluorescence imaging (800-900 nm) of a capillary tube containing iRFP713 immersed at various depths in 1% intralipid.

3. The selling point of the Ms. is that iRFPs provide much improved imaging penetration depth and contrast in the NIR-II window over their traditional NIR window. The authors argue in favor of this assumption based on capillary tube with Intralipid studies. This qualitative reasoning is not enough to support the *in vivo* imaging. The authors should better compare all the *in vivo* imaging of iRFPs under the NIR-I/II regions using both silicon and InGaAs cameras simultaneously, further helpful in determining the advantage of fluorescent protein assisted NIR-II imaging.

Response: We are grateful that the Reviewer pointed out a critical drawback of our previous manuscript. Following your instruction, we conducted *in vivo* imaging of the iRFP713^{flox/flox}; Alb-Cre, iRFP713^{flox/flox}; E2A-Cre, and iRFP713^{flox/flox}; Pdx1-Cre mice within 800-900 nm (by GA1280 camera and iXon Ultra 897 camera) and beyond 900 nm (by SW640 camera) to directly compare the intravital imaging quality, and the results have been integrated into Supplementary Fig. 15e. To quantitatively evaluate the image quality, we also measured the coefficient of variation (CV) value, and the data clearly showed that the images recorded by the InGaAs camera (>900 nm) possesses the highest CV values in all models, indicating the best image quality. Thus, we believe that the NIR-II bioimaging of iRFP713 using the InGaAs camera has advantages over the NIR-I bioimaging using the silicon camera.

Supplementary Figure 15e. NIR-I fluorescence imaging using the GA1280 camera (left) and iXon Ultra 897 camera (EMCCD; middle) and NIR-II fluorescence imaging using the SW640 camera (right).

4. I believe they need to do a set of important control studies. The authors might want to provide signal to background ratios (SBR) of current NIR-II imaging outcome, and compare the SBR with NIR-I data of the same *in vivo* models collected from silicon camera.

Response: Thanks for the valuable suggestion. We admit that NIR-I/II imaging comparison on the same mouse model was absent in the previous version, which is indeed important for our studies. As mentioned above, we have performed the *in vivo* imaging under NIR-I/II regions (Supplementary Fig.15e). After imaging, **area signal to background ratios (SBR)** of both NIR-I and NIR-II intravital images were evaluated. The ratio of the mean intensities of the signal area to those of the background area was calculated as the area SBR value of each image. As displayed in **Supplementary Table 1**, the area SBR of NIR-II (SW640 group) was significantly higher than that of NIR-I (GA1280 and iXon Ultra 897 groups) in all models, demonstrating the advantages of iRFP713-assisted NIR-II imaging. Combined with the result of calculated CV values in Supplementary Fig.15e, we therefore conclude that NIR-II imaging of iRFP713 possesses better imaging quality than the NIR-I.

Supplementary Table 1. The calculated area SBR values.

Area SBR values	GA1280	iXon Ultra 897	SW640
iRFP713 ^{flox/flox} ; Alb-Cre	3.07	21.62	213.99
iRFP713 ^{flox/flox} ; E2A-Cre	2.33	13.32	72.31
iRFP713 ^{flox/flox} ; Pdx1-Cre	2.95	17.83	47.07

5. The molecular biology data is relatively insufficient and weak. Could you shed light on the effect of the long-term fluorescence stability of transgenic model? Due to the lack of appropriate evidences and data, the statement of fluorescence up to 15 months is not convincing.

Response: In this study, we adopted the CRISPR-Cas9 technology to generate iRFP713 transgenic mice, allowing long-term stable expression of iRFP713. As expected, we did observe its fluorescence up to 15 months. To respond to the Reviewer's concern, we have evaluated the long-term stability of iRFP713 gene and protein expression in our transgenic mice. As shown in **Supplementary Fig. 14**, there were no significant differences of iRFP713 mRNA/protein expression level in young (8 weeks) and old (17 months) iRFP713^{flox/flox}; EIIa-Cre mice, as well as in iRFP713^{flox/flox}; Alb-Cre mice and iRFP713^{flox/flox}; Pdx1-Cre mice. Based on these molecular biology data, we firmly believe that iRFP713 fluorescence is stable for more than 15 months. We have added these data in DISCUSSION section of the revised manuscript (**Page 11, line 17-19**), as following:

“, and in our case, the transgenic mice had stable iRFP713 gene and protein expression for at least 15 months (Supplementary Fig. 14).”

Supplementary Figure 14. Long-term stability of iRFP713 gene and protein expression in different transgenic mouse models.

6. The NIR-II quantum yield (QY) is very important to evaluate the imaging ability of a contrast agent. A fluorophore with very low QY would fail to provide significantly improved imaging quality even under the NIR-II imaging window. The longer exposure time would increase the background signal.

Response: We admit that the quantum yield (QY) of the fluorophore is extremely important for fluorescence bioimaging. To address the Reviewer's concern, we tested the absolute QY of the iRFP713 using the absolute PL quantum yield spectrometer (Quantaury-QY, Hamamatsu Photonics) and calculated the NIR-II QY (beyond 900 nm) as only ~0.33%, which is indeed a little low. However, we could still achieve good performance of iRFP713-assisted *in vivo* imaging in the NIR-II window, even using very short exposure times (just tens of milliseconds, video rate) and low excitation power density (tens of mW cm⁻², the photothermal damage is negligible). We think the reason is that the exposure time and excitation power density of imaging rely on the photoluminescence brightness of the fluorophore, which is determined by the product of absorption coefficient and QY. As reported, the molar absorption coefficient of iRFP713 protein at 690 nm (the excitation wavelength in this study) is 98000 M⁻¹cm⁻¹ (*Nature Methods*, 2013, 10(8): 751-754), much larger than that of most existing fluorophores. The product of molar absorption coefficient and NIR-II QY can be calculated as ~3.2 [$\times 10^2$ M⁻¹cm⁻¹], while one reported excellent NIR-II dye (IR-FTAP) possesses the product of ~2.6 [$\times 10^2$ M⁻¹cm⁻¹] (*J. Am. Chem. Soc.* 2018, 140, 5, 1715–1724, ESI highly cited paper). Therefore, we think the photoluminescence brightness of iRFP713 beyond 900 nm is actually very high.

In addition, we agree with the review that the longer exposure time would increase the background signal. However, in our study, the brightness of iRFP713 is already high enough, and the long exposure time, which would also reduce the temporal resolution of the imaging, is not necessary. We have added these details in RESULTS section of the revised manuscript (Page 5, line 16-20), as following:

“For example, we tested the absolute QY of iRFP713 using an absolute PL quantum yield spectrometer and calculated the NIR-II QY (beyond 900 nm) as ~0.33%. Although the QY is not very high, the molar absorption coefficient of iRFP713 protein at the main peak is as high as 98000 M⁻¹cm⁻¹²⁶, leading to a strong NIR-II emission.”

7. *The text of the Ms. needs to be revised in order to correct for typos, or to avoid awkward phrasing.*

Response: Thanks for your kind reminding. We have polished our manuscript carefully and corrected all the wrong typos, please see the revised manuscript for details.

8. *The number of significant digits is not consistent in the manuscript, for instance, “191.0% ± 34.98% and 218.3% ± 53.8%” in Page 8.*

Response: We feel sorry for the mistake and correct “34.98%” to “35.0%” (Page 8, line 24). Additionally, we have double checked the number of significant digits in the revised manuscript carefully.

9. *Imaging conditions are missing in the Figure legend Figure 2. It should be mentioned, otherwise it is confusing to the readers.*

Response: As you suggested, we added the imaging exposure times in Fig. 2 (Page 16, line 6): “Exposure times: 35 ms (b, c) and 20 ms (d)”.

Additionally, to make it more rigorous, we have added all the imaging conditions in the revised manuscript, as following:

Figure 1 (Page 14, line 1-2): “Exposure times: (NIR-I) 0 mm and 2 mm: 10 ms; 4 mm: 80 ms; 6 mm: 100 ms. (NIR-II) 0 mm and 4 mm: 10 ms; 2 mm: 5 ms; 6 mm: 100 ms.”.

Figure 3 (Page 18, line 15-16): “Exposure times: 15 ms (b, e, f) and 10 ms (d)”.

Figure 4 (Page 20, line 10): “Exposure times: 20 ms (b, d, f) and 10 ms (c)”.

Supplementary Figure 1 (SI, Page 2, line 8): “Exposure times: 10 ms”.

Supplementary Figure 3 (SI, Page 4, line 10): “Exposure times: 50 ms”.

Supplementary Figure 5 (SI, Page 6, line 12): “Exposure times: 30 ms (b, c)”.

Supplementary Figure 6 (SI, Page 7, line 8): “Exposure times: 15 ms”.

Supplementary Figure 9 (SI, Page 10, line 12): “Exposure times: 15 ms”.

Supplementary Figure 12 (SI, Page 13, line 8): “Exposure times: 10 ms (a, left), 50 ms (a, middle) and 20 ms (a, right)”.

Supplementary Figure 13 (SI, Page 14, line 11): “Exposure times: 15 ms”.

Reference

1. Feng, Z. et al. Perfecting and extending the near-infrared imaging window. *Light, Science & Applications* **10**, 197 (2021).
2. Carr, J.A. et al. Absorption by water increases fluorescence image contrast of biological tissue in the shortwave infrared. *Proceedings of the National Academy of Sciences of the United States of America* **115**, 9080-9085 (2018).
3. Shcherbakova, D.M. & Verkhusha, V.V. Near-infrared fluorescent proteins for multicolor in vivo imaging. *Nature Methods* **10**, 751-754 (2013).
4. Yang, Q. et al. Donor Engineering for NIR-II Molecular Fluorophores with Enhanced Fluorescent Performance. *Journal of the American Chemical Society* **140**, 1715-1724 (2018).

Reviewer #2 (Remarks to the Author):

In this manuscript entitled “Long-term monitoring of intravital biological process using fluorescent protein assisted NIR-II imaging”, the authors tested non-invasive imaging using NIR fluorescent protein iRFP-expressing mice with longer wavelength 900-1880 nm (NIR-II).

Non-invasive imaging of living mice using NIR fluorescence is thought to be very powerful methods but spread of the technique is not enough so far, thus the data of this manuscript are very useful for many researchers.

Therefore, this reviewer think that it is very important that experimental details should be described sufficiently and precisely in this manuscript.

The experimental strategies are well designed and overall the data are clearly presented but in several points the descriptions are insufficient or not suitable.

Response: We are happy to see the Reviewer’s appreciation to non-invasive imaging of living mice and our work. According to your suggestion, we have provided detailed information and descriptions in the revised manuscript, including the procedure of mouse model construction, the guide RNA sequence, imaging conditions, images processing method, calculated scattering data, and so on. Please see the following detailed responses.

Major points

1) *Because they used CRISPR/Cas9 system to establish Tg mice, they should show the sequence of the guide RNA to reveal clearly the target site on the mouse genome.*

Response: The guide RNA sequence is: TGGGCGAGAAATGTGTCCTG, which targets to intron 1 of the Rosa26 locus at position +113025988 of chromosome 6. This information has been added in the METHODS section of the revised manuscript (Page 22, line 21-26), as following:

“In brief, the donor vector containing the “CAG promoter-loxP-3xpolyA-loxP-Kozak-iRFP713-polyA” cassette was constructed and subsequently inserted into intron 1 of the Rosa26 gene at position +113025988 of mouse chromosome 6. To generate targeted conditional knockin offspring, the guide RNA for mouse Rosa26 (matching forward strand of gene; TGGGCGAGAAATGTGTCCTG), Cas9 mRNA, and the target donor vector were coinjected into fertilized mouse eggs.”

2) *Because the intensity of NIR-II fluorescence is thought to be quite low, the exposure times should be shown and image adjustment method should be described in detail.*

Response: As required, we have added the description of exposure times (most were just tens of milliseconds) wherever needed in the revised manuscript, as following:

Figure 1 (Page 14, line 1-2): “Exposure times: (NIR-I) 0 mm and 2 mm: 10 ms; 4 mm: 80 ms; 6 mm: 100 ms. (NIR-II) 0 mm and 4 mm: 10 ms; 2 mm: 5 ms; 6 mm: 100 ms.”.

Figure 2 (Page 16, line 6): “Exposure times: 35 ms (b, c) and 20 ms (d)”.

Figure 3 (Page 18, line 15-16): “Exposure times: 15 ms (b, e, f) and 10 ms (d)”.

Figure 4 (Page 20, line 10): “Exposure times: 20 ms (b, d, f) and 10 ms (c)”.

Supplementary Figure 1 (SI, Page 2, line 8): “Exposure times: 10 ms”.

Supplementary Figure 3 (SI, Page 4, line 10): “Exposure times: 50 ms”.

Supplementary Figure 5 (SI, Page 6, line 12): “Exposure times: 30 ms (b, c)”.

Supplementary Figure 6 (SI, Page 7, line 8): “Exposure times: 15 ms”.

Supplementary Figure 9 (SI, Page 10, line 12): “Exposure times: 15 ms”.

Supplementary Figure 12 (SI, Page 13, line 8): “Exposure times: 10 ms (a, left), 50 ms (a, middle) and 20 ms (a, right)”.

Supplementary Figure 13 (SI, Page 14, line 11): “Exposure times: 15 ms”.

Image adjustment method has been described in detail in METHODS section (Page 24, line 23-32), as following:

“Image processing. Quantitative analysis of the fluorescent images was performed using ImageJ software (Version 1.6.0, National Institutes of Health, USA) based on the measurement of mean signal intensity in the manually selected regions of interest. All images were processed using the same settings within a test for both the control and experimental groups. For NIR-II imaging of the liver, the grayscale image sequence was taken and binarized by the same threshold in image preprocessing. According to the binarized image, image segmentation was performed with light as the signal area and dark as the background area. The sum of the value on each pixel of the bright area was considered to be the total fluorescence intensity, as was the intensity of the background signal. Then, the signal-to-background ratio (SBR) was calculated.”

3) In Fig1 some important experimental descriptions are insufficient.

Fig1a: this reviewer can not understand the reason why the 3D structure of iRFP is needed for this manuscript.

Response: The Reviewer is right, and the 3D structure is not necessary at all. We have deleted this structure in the revised manuscript.

Fig1c: Wavelength used for excitation should be shown.

The authors show the spectra in 600 - 1400 nm and detail in 600 - 800 is not clear. The data should be showed in 600 - 1000 nm

Response: Thank you very much for this excellent suggestion. Following your instruction, we have adjusted the abscissa of the original Fig. 1c to 600-1000 nm to show more details in 600-800 nm. Additionally, to make the reading easier, we split the original Fig. 1c into new Fig. 1b and Fig. 1c in the revised manuscript.

Original Figure 1c. Normalized fluorescence emission spectra of five purified iRFPs.

Revised Figure 1b-c. (b) Normalized fluorescence emission spectra of five purified iRFPs. (c) Emission spectra of iRFPs in the range of 900-1300 nm.

Fig1d: the intensity of iRFP670 seems to be too high because excitation rate is only 10%(Fig1b) in 690 nm and emission is very low (Fig1c). The authors should explain this contradiction.

Response: Although the excitation rate was only ~10% as presented in the normalized absorption spectrum in the original Fig. 1b (now revised as Fig. 1a), the molar absorption coefficient of iRFP670 at the main peak is as high as $114000 \text{ M}^{-1}\text{cm}^{-1}$ (*Nature Methods*, 2013, 10(8): 751-754). Therefore, the molar absorption coefficient at 690 nm was at least $\sim 11400 \text{ M}^{-1}\text{cm}^{-1}$, thus resulting its moderate fluorescence intensity.

To address the Reviewer's concern, we have supplemented certain explanation in RESULTS section of the revised manuscript (Page 5, line 13-16), as following:

“It should be noted that the iRFPs possess high molar absorption coefficients²⁶. Thus, they showed good fluorescence performance, as shown in Fig. 1d, even though the light absorption proportion at 690 nm of some proteins was relatively low, as shown in the normalized absorption spectra in Fig. 1a.”

Fig1f: this reviewer think that the data clearly show the advantage of NIR-II than NIR-I in light scattering. However, in upper data, intensity remains to be high even in deeper conditions (4 mm, 6 mm) but scattering becomes clearly higher. Usually the relationship between intensity and scattering is tradeoff and scattering becomes higher, intensity becomes lower like in lower data.

The authors should explain this point.

The quantitative data of scattering (for example CV value) might be very useful for many researchers and if it is possible please show calculated data of the scattering.

Response: We are sorry to make the Reviewer confused due to not providing necessary imaging information and data. We believe the different exposure times in our experiments (the larger imaging depth, the longer exposure time) caused the different integrated intensities so that the signal and scattering background both became stronger with increasing depth. To avoid ambiguity, we added the exposure times in the figure caption (Fig. 1f).

Following your kind suggestion, we measured the coefficient of variation (CV) values of the *in vitro* intralipid phantom images and added them in revised Fig. 1f. It can be found that the CV values of the NIR-II images were higher than those of NIR-I images at the depth of 2 mm, 4 mm, and 6 mm, indicating the NIR-II images possessed weaker scattering background (negatively correlated with CV value). Besides, we believe the *in vivo* imaging comparison would be more convincing about the superiority of the NIR-II imaging window. Aiming to quantitatively evaluate the intravital imaging quality, we thus measured the CV value of intravital fluorescence images. Consistent with *in vitro* results, CV values of the images beyond 900 nm were higher than those of 800-900 nm images in all *in vivo* models (Supplementary Fig. 15e). Taken together, we concluded that the scattering background of iRFP713 imaging in the NIR-II window was efficiently inhibited.

We have integrated the aforementioned information into the revised manuscript, including the CV values of *in vitro* intralipid phantom images in the RESULTS (Page 6, line 4-5) and that of *in vivo* models in DISCUSSION (Page 11, line 34 and Page 12, line 1-5) sections, as following:

Page 6, line 4-5: “The quantitative results of the coefficient of variation (CV) values further indicated the potential of iRFP713 for NIR-II fluorescence imaging.”

Page 11, line 34 and Page 12, line 1-5: “The *in vitro* intralipid phantom imaging and *in vivo* imaging of transgenic mice (Supplementary Fig. 15 and Supplementary Table 1) using the above three cameras showed that images recorded in the NIR-II window had the best imaging quality, whether the EMCCD or vis-NIR camera was used for NIR-I detection. Thus, we believe that the excellent imaging performance in the NIR-II window should be credited to the window itself.”

Figure 1f. NIR fluorescence imaging of a capillary tube containing iRFP713 immersed at various depths in 1% intralipid. Exposure times: (NIR-I) 0 mm and 2 mm: 10 ms; 4 mm: 80 ms; 6 mm: 100 ms. (NIR-II) 0 mm and 4 mm: 10 ms; 2 mm: 5 ms; 6 mm: 100 ms.

Supplementary Figure 15e. The NIR-I fluorescence (800-900 nm) images using the GA1280 camera (left) and iXon Ultra 897 camera (middle) and the NIR-II fluorescence (>900 nm) images using the SW640 camera (right).

4) Normal mouse food and milk from mother mouse usually show high intensity of NIR autofluorescence and some low NIR-fluorescence mouse foods are commercially available. Change of mouse food before taking photo is very important experimental technique to avoid the effects of autofluorescence.

In Fig 2 not only iRFP fluorescence but also milk autofluorescence seems to be overlapped.

In Fig3 and Fig4 no autofluorescence of food is detected. If the authors change mouse food before taking photo, they should make clear that point.

Response: We appreciate the Reviewer's advice and friendly reminding. Indeed, normal mouse food and mother milk have been reported to have NIR autofluorescence. However, since the autofluorescence intensity is always relatively low, under the laser excitation of power density at tens of mW cm^{-2} (in this work), enough exposure times (such as over 100 ms) are needed to make the weak autofluorescence detectable by the camera. Notably, the exposure times used in this work were just 20/35 ms in Fig 2, 10/15 ms in Fig 3 and 10/20 ms in Fig 4. Under this imaging condition, autofluorescence of food and mother milk was undetectable even if taking a close look at the control mouse in these figures.

We did not change the mouse food before and after *in vivo* imaging. Just to be clear, to establish APAP-induced liver injury model in Fig.3 and cerulein-induced pancreatitis model in Fig.4, the mouse food was removed for 12 hours before APAP and cerulein injections, and was subsequently returned after injections. The information has been described in METHODS section (Page 23, line 11-16 and Page 24, line 6-13).

Page 23, line 11-16: "To establish a liver regeneration model induced by acetaminophen (APAP) (Sigma Aldrich), 2- to 3-month-old male iRFP713^{flx/flx}; Alb-Cre mice were starved for 12 hours with free access to water and injected intraperitoneally with 250 mg/kg APAP dissolved in warm 0.9% saline at a concentration of 15 mg/mL. After IP injection of APAP, the mice were returned to normal food. NIR-II fluorescence imaging of mice was performed at 1 and 2 days after APAP treatment."

Page 24, line 6-13: "Cerulein-induced pancreatitis model. To establish an acute pancreatitis model, 8-week-old male iRFP713^{flx/flx}; Pdx1-Cre mice were treated with cerulein (Sigma Aldrich). Briefly, before the experiment, the mice were fasted overnight and allowed water ad libitum. Mice were then injected intraperitoneally with 50 $\mu\text{g/kg}$ cerulein dissolved in 0.9% saline in a volume of 100 μL . Cerulein injections were undertaken in hourly intervals for up to 7 injections. After injections, the mice were returned to the normal diet. NIR-II fluorescence *in vivo* imaging of mice was conducted before the first injection and 24, 48 and 72 hours afterward."

Minor point

The previous report establishing EIIa-cre mouse should be cited.

Response: Following the Reviewer's suggestion, we have cited the reference (*Proceedings of the National Academy of Sciences, 1996, 93(12): 5860-5865*) in the revised manuscript (Page 6, line 34).

Reference

1. Shcherbakova, D.M. & Verkhusha, V.V. Near-infrared fluorescent proteins for multicolor in vivo imaging. *Nature Methods* **10**, 751-754 (2013).
2. Lakso, M. et al. Efficient in vivo manipulation of mouse genomic sequences at the zygote stage. *Proceedings of the National Academy of Sciences of the United States of America* **93**, 5860-5865 (1996).

Reviewer #3 (Remarks to the Author):

General Comments: In this manuscript, the authors have described a novel iRFP713 fluorescent protein based in vivo imaging technology. They have shown that iRFP can be used to track change in organ size (liver and pancreas) in models of regeneration. Overall, the studies are well conducted and interesting. However, significant details of the experiments and the actual cellular localization and its effects on the cells of the new iRFP713 protein are missing.

Response: We appreciate that the Reviewer positively evaluated our work, and critically pointed out the issues needed to be improved. Following the Reviewer's instruction, we have provided essential experimental details, and conducted experiments to determine the cellular localization of the iRFP713 protein and its effects on the cells. Please see below for more information.

Detailed Comments:

1. *Why is iRFP713 expression so much higher in liver than other tissues, especially given the cre used is not liver specific?*

Response: First of all, we should admit that it has been often observed that an exogenous protein highly expresses in the liver, for example:

Representative images of the tdTomato fluorescence in the AAV-transduced reporter mice (*Nature communications*, 2020, 11(1): 1-11, Fig. 4d.)

iRFP expression in organs

Next, we harvested most organs for imaging, including the brain, heart, liver, kidney, spleen, lung, pancreas, testis, bone, testis, thymus, and adipose tissue. In both mouse lines, the fluorescence intensity was similar among almost organs, but was higher in the lung, pancreas and especially liver. The fluorescence intensity in most organs from the 846 line was dimmer than 867 line (Figs. 2 A-L). These data suggest that the fluorescence intensity of line 867 is higher than line 846. Taken together, iRFP expression driven by the CAG promoter was observed in various organs.

Whole-body and organs imaging of iRFP mice. (*Experimental Animals*, 2014, 63(3): 311-319, Fig. 4d and Fig. 4e.)

In our case, the iRFP713^{flx/flx}; EIIa-Cre mice carry the adenovirus EIIa promoter (ubiquitous, not liver specific). Conceptually, EIIa promoter directs Cre recombinase expression in early mouse embryo, thus evokes ubiquitous expression of downstream target genes (*Proceedings of the National Academy of Sciences*, 1996, 93(12): 5860-5865). However, it should be noted that the Cre expression and Cre-mediated recombination efficiency could be variable in different kinds of cell types (*Proceedings of the National Academy of Sciences*, 2017, 114(3): 498-503), thus resulting in differential iRFP713 expression across organs. In addition, the production of iRFP713 protein depends not only on sufficient levels of Cre recombinase to excise the loxP-flanked STOP cassette but also on protein biosynthesis. As we know that liver is the largest and a vital organ that has extensive metabolic and synthetic functions, high Cre recombination efficiency and iRFP713 synthesis capacity may account for the high iRFP713 expression in this organ.

2. Which cells in the liver express iRFP713?

Response: This is an interesting question. The liver is comprised of four basic cell types: hepatocytes (account for >75% of the liver volume) and other non-parenchymal cells, including endothelial cells, Kupffer cells and stellate cells (*Developmental cell*, 2010, 18(2): 175-189; *Cell metabolism*, 2014, 20(1): 85-102; *Nature communications*, 2015, 6(1): 1-12). In this study, we employed two transgenic mice (iRFP713^{flx/flx}; EIIa-Cre and iRFP713^{flx/flx}; Alb-Cre) whose liver cells carried iRFP713 under the control of EIIa promoter and albumin (Alb) promoter.

1) For EIIa-Cre promoter, previous studies demonstrated that EIIa-Cre mice yielded a proportion of progeny showing mosaic Cre deletion patterns, since Cre recombinase was not expressed until fertilization occurs (*Proceedings of the National Academy of Sciences*, 1996, 93(12): 5860-5865; *Genesis*, 2001, 30(1): 1-6). Besides, Heffner *et al* reported that paternally inherited EIIa-cre displayed mosaic recombination in liver (*Nature communications*, 2012, 3(1): 1-9. **Please find the figure below**). Consistently, our data also showed mosaic expression of iRFP713 in liver cells of the EIIa-Cre mice (both parenchymal cells and non-parenchymal cells) (**Supplementary Fig. 5c**).

2) For Alb Cre promoter, it is reported that albumin (Alb) gene is only expressed in adult hepatocytes (*Genesis*, 2000, 26(2): 151-153; *The Journal of clinical investigation*, 2011, 121(12): 4850-4860), thus the Cre recombinase derived by this promoter also expressed in this type of cell, resulting a hepatocyte-specific expression of iRFP713 (**Fig. 3b**).

Mosaic expression pattern of Cre-mediated β -gal activity in liver of EIIa-Cre mice (*Nature communications*, 2012, 3(1): 1-9, Fig. 4e)

Supplementary Figure 5c. NIR-II fluorescence images of isolated livers from iRFP713^{lox/lox}, EIIa-Cre mice.

Figure 3b. NIR-II fluorescence images of isolated livers from iRFP713^{lox/lox}, Alb-Cre mice.

3. The experiments shown in Fig 3 on PHX are interesting. However, more information is needed to rule out possibility that iRFP expression changes kinetic of proliferation and pathways involved in regeneration. Determine cell proliferation using either PCNA or Ki-67, and mRNA of Cyclin D1 is important. A correlation of these markers with iRFP fluorescence can be a good additional measure.

Response: The Reviewer raised a key question about the effect of iRFP713 expression on cell proliferation during partial hepatectomy (PHX). To address your concern, we first determined the expression of cyclin D1. As shown in Supplementary Fig. 8a-c, its mRNA and protein levels were highly upregulated in both liver-specific mice (iRFP713^{flox/flox}; Alb-Cre) and control mice (iRFP713^{flox/flox}) after PHX; however, there were no differences between groups. Besides, in line with the results of cyclin D1, no significant differences were detected for Ki-67 immunostaining between the two groups (Supplementary Fig. 8d-e). Based on these data, we thus concluded that iRFP713 expression did not alter the proliferation progress after PHX.

We further examined the correlation of iRFP713 fluorescence with proliferation makers. As shown in Supplementary Fig. 8f-g, there were positive correlations between iRFP713 fluorescence intensity and cyclin D1 protein expression ($R^2 = 0.7149$, $p < 0.01$), and Ki-67 staining ($R^2 = 0.6455$, $p < 0.01$), indicating that iRFP713 fluorescence positively correlates with hepatocyte proliferation during PHX.

We have integrated these data into the RESULTS section of the revised manuscript (Page 8, line 17-20), as following:

“Notably, our results revealed that increased iRFP713 expression did not affect liver proliferation, as demonstrated by unaltered cyclin D1 and Ki-67 expression (Supplementary Fig. 8a-e), but was positively correlated with hepatocyte proliferation after PHX (Supplementary Fig. 8f-g).”

Supplementary Figure 8a-c. Liver mRNA (a) and protein levels (b, c) of cyclin D1 on different days after PHX.

Supplementary Figure 8d-e. Ki-67 IHC staining on different days after PHX.

Supplementary Figure 8f-g. The correlation between iRFP713 fluorescence intensity and cyclin D1 expression level (f), and Ki-67 staining (g).

4. To compare ICG and iRFP, parameters of proliferation and injury should be measured in both groups at the same time.

Response: Following the Reviewer's instruction, we assessed liver proliferation and injury simultaneously in both iRFP713^{fllox/fllox}; Alb-Cre mice and ICG-administrated C57BL/6 mice. Our results showed that there were no obvious differences in cyclin D1 mRNA and protein expression levels between iRFP713 and ICG groups at one day after PHX (Supplementary Fig. 10a-b). Meanwhile, no significant changes were observed in the Ki-67 staining, suggesting that the two groups of mice did not differ in liver proliferation (Supplementary Fig. 10c).

We further evaluated the extent of liver injury by TUNEL assay, and found that the number of TUNEL-positive cells in ICG group was notably higher compared with the iRFP713 group (Supplementary Fig. 10d). This finding was consistent with our previous result that ICG treatment exhibited hepatotoxicity with elevated ALT, AST, ALP and TBIL levels on day 1 post-PHX (Supplementary Fig. 9b). Taken together, we concluded that ICG administration resulted in severe liver injury, with no difference in proliferation compared to iRFP713.

To highlight the advantages of iRFP713 over ICG, we have added these data in the RESULTS section (Page 9, line 6-9), and provided the detailed experimental protocol in the METHODS section (Page 23, line 18-23), as following:

Page 9, line 6-9: "More importantly, although ICG did not alter hepatic proliferation (Supplementary Fig. 10a-c), it triggered severe hepatocyte apoptosis, as indicated by the increased number of TUNEL-positive cells (Supplementary Fig. 10d)."

Page 23, line 18-23: "Proliferation and apoptosis assays. Tissue samples from mice of the corresponding genotypes were isolated, weighed and divided into aliquots for RT-qPCR, western blot, hematoxylin and eosin (HE) staining, and immunohistochemical (IHC) staining. Cell proliferation was evaluated by Ki-67 IHC staining according to the standard procedure. A TUNEL assay was performed to investigate cell apoptosis according to the manufacturer's manual (Elabscience)."

Supplementary Figure 10a-b. Liver mRNA (a) and protein (b) levels of cyclin D1 in iRFP713^{fllox/fllox}; Alb-Cre mice and ICG-administrated C57BL/6 mice at one day after PHX.

Supplementary Figure 10c. Ki-67 IHC staining on liver sections of iRFP713^{flx/flx}; Alb-Cre mice and ICG-administrated C57BL/6 mice at one day after PHX.

Supplementary Figure 10d. TUNEL assay on liver sections of iRFP713^{flx/flx}; Alb-Cre mice and ICG-administrated C57BL/6 mice at one day after PHX.

Supplementary Figure 9b. Four main hepatic function indexes of ICG-treated mice at one day after PHX.

5. The data on acetaminophen toxicity model need lot more work. It is known that APAP overdose does not affect liver weight to any appreciable degree. The injury is mainly centrilobular (zone 3) in the liver and hepatocytes around the necrotic zone regenerate. It is not clear why iRFP declines after APAP administration. It may be due to the fact that albumin promoter doesn't work as well or may be due to overall cell death. This needs to be investigated. More experimental details such as the exact protocol used (fasted non-fasted, was the food returned after APAP treatment) and profiles of liver injury (did APAP actually produce injury or not) and recovery should be provided.

Response: We appreciate the Reviewer for proposing two possibilities to explain the attenuated iRFP713 fluorescence after APAP. To test the first possibility, we examined liver iRFP713 gene and protein expression, which would be declined when Alb promoter did not work. As shown in Supplementary Fig. 12b-c, APAP administration did not alter iRFP713 expression compared to the control, demonstrating that the declined hepatic fluorescence was not caused by impaired Alb promoter-dependent iRFP713 expression.

To examine the second possibility, we evaluated the liver cell death after APAP treatment. Our HE and TUNEL staining showed large areas of necrosis around centrilobular at one day post APAP challenge (Supplementary Fig. 12d, left and middle). Meanwhile, IHC staining showed that iRFP713 was concentrated in non-necrotic regions, whereas was nearly absent within necrotic zone (Supplementary Fig. 12d, right), suggesting that APAP-induced hepatic necrosis remarkably attenuated iRFP713 concentration in liver. Based on these data, we therefore conclude that the reduction of iRFP713 fluorescence is caused by the hepatocyte death. We have integrated these new obtained data into Supplementary Figure 12, and rewritten this part in the RESULTS section of the revised manuscript (Page 9, line 14-16), as following:

“iRFP713-assisted imaging revealed that the hepatic fluorescence intensity declined by 62.8% at day 1 (Supplementary Fig. 12a and Fig. 3f) due to the APAP-induced death of hepatocytes (Supplementary Fig. 12b-d)”

In addition, we examined the liver injury and recovery after APAP exposure. As shown in Supplementary Fig. 11, in distinct injury phase (day 1 post APAP injection), there was typical centrilobular necrosis as indicated by cellular vacuolization, cell swelling, and nuclear disintegration. In resolution phase (day 5 post APAP injection), the necrotic area was reduced, centrilobular hepatocellular necrosis was nearly resolved, and the liver appeared normal. Thus, this result illuminated the process of liver injury and recovery after APAP challenge. We have included these data to Supplementary Figure 11.

For the procedure of APAP administration, mice were starved for 12 hours with free access to water before APAP exposure, and were returned to food after APAP. We have included the detailed description in the METHODS section of the revised manuscript (Page 23, line 11-16), as following:

“To establish a liver regeneration model induced by acetaminophen (APAP) (Sigma Aldrich),

2- to 3-month-old male *iRFP713^{lox/lox}*; Alb-Cre mice were starved for 12 hours with free access to water and injected intraperitoneally with 250 mg/kg APAP dissolved in warm 0.9% saline at a concentration of 15 mg/mL. After IP injection of APAP, the mice were returned to normal food. NIR-II fluorescence imaging of mice was performed at 1 and 2 days after APAP treatment.”

Supplementary Figure 12b-c. iRFP713 gene (b) and (c) protein levels in liver of *iRFP713^{lox/lox}*; Alb-Cre mice after APAP treatment.

Supplementary Figure 12d. Representative HE staining (left), TUNEL immunofluorescence staining (middle) and IHC staining of iRFP713 (right) in liver sections from *iRFP713^{lox/lox}*; Alb-Cre mice after APAP treatment.

Supplementary Figure 11. HE staining of liver isolated from *iRFP713^{lox/lox}*; Alb-Cre mice after APAP treatment.

Reference

1. Wu, J. et al. A non-invasive far-red light-induced split-Cre recombinase system for controllable genome engineering in mice. *Nature communications* **11**, 3708 (2020).
2. Tran, M.T. et al. In vivo image analysis using iRFP transgenic mice. *Experimental animals* **63**, 311-319 (2014).
3. Lakso, M. et al. Efficient in vivo manipulation of mouse genomic sequences at the zygote stage. *Proceedings of the National Academy of Sciences of the United States of America* **93**, 5860-5865 (1996).
4. Zhang, G. et al. p53 pathway is involved in cell competition during mouse embryogenesis. *Proceedings of the National Academy of Sciences of the United States of America* **114**, 498-503 (2017).
5. Si-Tayeb, K., Lemaigre, F.P. & Duncan, S.A. Organogenesis and development of the liver. *Developmental cell* **18**, 175-189 (2010).
6. Gurzov, E.N. et al. Hepatic oxidative stress promotes insulin-STAT-5 signaling and obesity by inactivating protein tyrosine phosphatase N2. *Cell metabolism* **20**, 85-102 (2014).
7. Bhate, A. et al. ESRP2 controls an adult splicing programme in hepatocytes to support postnatal liver maturation. *Nature communications* **6**, 8768 (2015).
8. Xu, X. et al. Direct removal in the mouse of a floxed neo gene from a three-loxP conditional knockout allele by two novel approaches. *Genesis* **30**, 1-6 (2001).
9. Heffner, C.S. et al. Supporting conditional mouse mutagenesis with a comprehensive cre characterization resource. *Nature communications* **3**, 1218 (2012).
10. Kellendonk, C., Opherk, C., Anlag, K., Schutz, G. & Tronche, F. Hepatocyte-specific expression of Cre recombinase. *Genesis* **26**, 151-153 (2000).
11. Malato, Y. et al. Fate tracing of mature hepatocytes in mouse liver homeostasis and regeneration. *The Journal of clinical investigation* **121**, 4850-4860 (2011).

Reviewers' Comments:

Reviewer #1:

Remarks to the Author:

The authors have adequately addressed the comments from this reviewer and the other reviewer (mostly technical questions).

Reviewer #3:

Remarks to the Author:

No further comments.

Point-by-point response to reviewers' comments

Reviewer #1 (Remarks to the Author):

The authors have adequately addressed the comments from this reviewer and the other reviewer (mostly technical questions).

Response: We appreciate the encouragement for Reviewer #1.

Reviewer #3 (Remarks to the Author):

No further comments.

Response: We thank the kindness from Reviewer #3.